# Geometry-Calibrated DRO: Combating Over-Pessimism with Free Energy Implications

## Abstract

Machine learning algorithms minimizing average risk are susceptible to distributional shifts. Distributionally Robust Optimization (DRO) addresses this issue by optimizing the worst-case risk within an uncertainty set. However, DRO suffers from over-pessimism, leading to low-confidence predictions, poor parameter estimations as well as poor generalization. In this work, we conduct a theoretical analysis of a probable root cause of over-pessimism: excessive focus on noisy samples. To alleviate the impact of noise, we incorporate data geometry into calibration terms in DRO, resulting in our novel Geometry-Calibrated DRO (GCDRO) for regression. We establish that our risk objective aligns with the Helmholtz free energy in statistical physics, and this free-energy-based risk can extend to standard DRO methods. Leveraging gradient flow in Wasserstein space, we develop an approximate minimax optimization algorithm with a bounded error ratio and standard convergence rate and elucidate how our approach mitigates noisy sample effects. Comprehensive experiments confirm GCDRO's superiority over conventional DRO methods.

## 1 Introduction

Machine learning algorithms with empirical risk minimization (ERM) have been shown to perform poorly under distributional shifts, especially sub-population shifts where substantial data subsets are underrepresented in the average risk due to their small sample sizes. As an alternative, Distributionally Robust Optimization (DRO) (Namkoong and Duchi, 2017; Blanchet and Murthy, 2019; Blanchet et al., 2019a; Duchi and Namkoong, 2021; Zhai et al., 2021; Liu et al., 2022a; Gao and Kleywegt, 2022; Gao et al., 2022) aims to optimize against the worst-case risk distribution within a predefined uncertainty set. This uncertainty set is centered around the training distribution, and generalization performance can be guaranteed when the test distribution falls within this set.

However, DRO methods have been found to experience the over-pessimism problem in practice (Hu et al., 2018; Zhai et al., 2021) (*i.e.*, low-confidence predictions, poor parameter estimations, and generalization), recent studies have sought to address this issue. From the *uncertainty set perspective*, Blanchet et al. (2019b); Liu et al. (2022a,b) proposed data-driven methods to learn distance metrics from data. However, these approaches remain vulnerable to noisy samples, as demonstrated in Table 2. Recently, Słowik and Bottou (2022); Agarwal and Zhang (2022) observed that DRO may overly focus on sub-populations with higher noise levels, leading to suboptimal generalization. Consequently, from the *risk objective perspective*, they suggest incorporating calibration terms to mitigate this issue. Nevertheless, applicable calibration terms either require expert knowledge or are computationally intensive, and few practical algorithms have been proposed.

To devise a practical calibration term for DRO, we first aim to identify the root causes of over-pessimism, which we attribute to the excessive focus on noisy samples that frequently exhibit higher

prediction errors. For typical DRO methods (Namkoong and Duchi, 2017; Staib and Jegelka, 2019; Duchi and Namkoong, 2021; Liu et al., 2022b), based on a simple yet insightful linear example, we theoretically demonstrate that the variance of estimated parameters becomes substantially large when noisy samples have higher densities, in line with the empirical findings reported in (Zhai et al., 2021). Furthermore, we demonstrate that existing outlier-robust regression methods are not directly applicable for mitigating noisy samples in DRO scenarios where both noisy samples and distribution shifts coexist, highlighting the non-trivial nature of this problem.

In this work, inspired by the ideas in (Słowik and Bottou, 2022; Agarwal and Zhang, 2022), we design calibration terms, *i.e.*, total variation and entropy regularization, to prevent DRO from excessively focusing on random noisy samples. In conjunction with the Geometric Wasserstein uncertainty set (Liu et al., 2022b) utilized in our methods, these calibration terms effectively incorporate information from the data manifold, leading to improved regulation of the worst-case distribution in DRO. Specifically, during the optimization, the total variation term penalizes the variation of weighted prediction errors along the data manifold, preventing random noisy samples from gaining excessive densities. The entropy regularization term, also used in (Liu et al., 2022b), acts as a non-linear graph Laplacian operator that enforces the smoothness of the sample weights along the manifold. These calibration terms work together to render the worst-case distribution more *reasonable* for DRO, leading to our Geometry-Calibrated DRO (GCDRO) approach. We validate the effectiveness of our GCDRO on both simulation and real-world data.

Furthermore, from a statistical physics perspective, we demonstrate that our risk objective corresponds to the Helmholtz free energy, comprising three components: interaction energy, potential energy, and entropy. The free energy formulation generalizes typical DRO methods such as KL-DRO, $\chi^2$-DRO (Duchi and Namkoong, 2021), MMD-DRO (Staib and Jegelka, 2019) and GDRO (Liu et al., 2022b). This physical interpretation provides a novel perspective for understanding different DRO methods by drawing parallels between the worst-case distribution and the steady state in statistical physics, offering valuable insights. From the free energy point of view, our GCDRO *specifically addresses the interaction energy between samples to mitigate the effects of noisy samples*. Motivated by the study of the Fokker-Planck equation (FPE, Chow et al. (2017); Esposito et al. (2021)), through gradient flow in the Geometric Wasserstein space, we derive an approximate minimax algorithm with a bounded error ratio $e^{-CT_{in}}$ after $T_{in}$ inner-loop iterations and a convergence rate of $\mathcal{O}(1/\sqrt{T_{out}})$ after $T_{out}$ outer-loop iterations. Our optimization method supports any quadratic form of interaction energy, potentially paving the way for designing more effective calibration terms for DRO in the future.

## 2   Preliminaries: Noisy Samples Bring Over-Pessimism in DRO

**Notations.**   $X \in \mathcal{X}$ denotes the covariates, $Y \in \mathcal{Y}$ denotes the target, $f_\theta(\cdot) : \mathcal{X} \to \mathcal{Y}$ is the predictor parameterized by $\theta \in \Theta$. $\hat{P}_N$ denotes the empirical counterpart of distribution $P(X, Y)$ with $N$ samples, and $\mathbf{p} = (p_1, \ldots, p_N)^T \in \mathbb{R}_+^N$ is the probability vector. $[N] = \{1, 2, \ldots, N\}$ denotes the set of integers from 1 to $N$. The random variable of data points is denoted by $Z = (X, Y) \in \mathcal{Z}$. The random vector of $n$ dimension is denoted by $\vec{h}_n = (h_1, \ldots, h_n)^T$. $G_N = (V, E, W)$ denotes a finite weighted graph with $N$ nodes, where $V = [N]$ is the vertex set, $E$ is the edge set and $W = \{w_{ij}\}_{(i,j)\in E}$ is the weight matrix of the graph. And $(x)_+ = \max(x, 0)$.

Distributionally Robust Optimization (DRO) is formulated as:

$$\theta^*(P) = \arg\min_{\theta \in \Theta} \sup_{Q \in \mathcal{P}(P)} \mathbb{E}_Q[\ell(f_\theta(X), Y)] \tag{1}$$

where $\ell$ is the loss function (typically mean square error) and $\mathcal{P}(P) = \{Q : \text{Dist}(Q, P) \le \rho\}$ denotes the $\rho$-radius uncertainty ball around the distribution $P$. Different distance metrics derive different DRO methods, e.g., $f$-divergence DRO ($f$-DRO, Namkoong and Duchi (2017); Duchi and Namkoong (2021)) with the Cressie-Read family of Rényi divergence, Wasserstein DRO (WDRO, Sinha et al. (2018); Blanchet and Murthy (2019); Blanchet et al. (2019a,b)), MMD-DRO (Staib and Jegelka, 2019) with maximum mean discrepancy, and Geometric DRO (GDRO, Liu et al. (2022b)) with Geometric Wasserstein distance. Although DRO methods are designed to resist sub-population shifts, they have been observed to have poor generalization performances (Hu et al., 2018; Frogner et al., 2019; Słowik and Bottou, 2022) in practice, which is referred to as over-pessimism.

In this section, we identify one of the root causes of the over-pessimism of DRO: the *excessive focus on noisy samples with typically high prediction errors*.

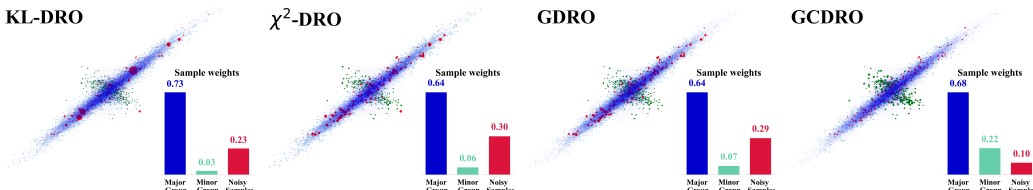

Figure 1: Visualizing the Worst-Case Distribution for Different DRO Methods: We show the data manifold and sample weights for each point, where blue points represent the major group, green ones represent the minor group, and red ones are noisy samples. The bars display the total sample weights of different groups, and the *original* group ratio is major (**93.1%**), minor (**4.9%**), (noisy **2%**).

- We showcase DRO methods' excessive focus on noisy samples in practice and reveal their probability densities are linked to high prediction errors in worst-case distributions.
- Through a simple yet insightful regression example, we prove that such a phenomenon leads to high estimation variances and subsequently poor generalization performance.
- We demonstrate that existing outlier-robust regression methods are not directly applicable for mitigating noisy samples in DRO scenarios, emphasizing the non-trivial nature of this problem.

**Problem Setting**   Given the *underlying* clean distribution $P_{clean} = (1-\alpha)P_{major} + \alpha P_{minor}, 0 < \alpha < \frac{1}{2}$, the **goal of DRO can be viewed as achieving good performance across all possible sub-populations** $P_{minor}$. Denote the observed contaminated training distribution by $P_{train}$. Based on Huber's $\epsilon$-contamination model (Huber, 1992), we formulate $P_{train}$ as:

$$P_{train} = (1-\epsilon)P_{clean} + \epsilon\tilde{Q} = \underbrace{(1-\epsilon)(1-\alpha)P_{major}}_{\text{major sub-population}} + \underbrace{(1-\epsilon)\alpha P_{minor}}_{\text{minor sub-population}} + \underbrace{\epsilon\tilde{Q}}_{\text{noisy sub-population}} , \quad (2)$$

where $\tilde{Q}$ is an arbitrary *noisy* distribution (typically with larger noise scale), $0 < \epsilon < \frac{1}{2}$ is the noise level. Note that the *minor sub-population could represent any distribution with a proportion of $\alpha$ in $P$*. However, we explicitly specify it here to emphasize the distinction between our setting and the traditional Huber's $\epsilon$-contaminated setting, as the latter does *not* take sub-population shifts into account.

**Empirical Observations.**   Following a typical regression setting (Duchi and Namkoong, 2021; Liu et al., 2022b), we demonstrate the worst-case distribution of KL-DRO, $\chi^2$-DRO, and GDRO in Figure 1, where the size of each point is proportional to its density. In this scenario, the underlying distribution $P$ comprises a known major sub-population (95%, blue points) and a minor sub-population (5%, green points). And the noise level $\epsilon$ in $P_{train}$ is 2%. DRO methods are expected to upweigh samples from minor sub-population to learn a model with uniform performances w.r.t. sub-populations. However, from Figure 1, we could observe that KL-DRO, $\chi^2$-DRO and GDRO excessively focus on noisy samples, resulting in a noise level 10 to 15 times larger than the original. This observation helps to explain their poor performance on this task (detailed results can be found in Table 2).

**Theoretical Analysis.**   To support our observations, we first analyze the worst distribution of KL-DRO, $\chi^2$-DRO and GDRO, shedding light on the underlying reasons for this phenomenon.

**Proposition 2.1** (Worst-case Distribution). *Let $\hat{Q}_N^* = (q_1^*, q_2^*, \ldots, q_N^*)^T \in \mathbb{R}_+^N$ denotes the worst-case distribution, and $\ell(f_\theta(x_i), y_i)$ (abbr. $\ell_i$) denotes the prediction error of sample $i \in [N]$. For different choices of $Dist(\cdot, \cdot)$ in $\mathcal{P}(P) = \{Q : Dist(Q, P) \leq \rho\}$, we have:*
- *KL-DRO: $q_i^*/q_j^* \propto \exp(\ell_i - \ell_j)$;*
- *GDRO's final state (gradient flow step $T \to \infty$): $q_i^*/q_j^* \propto \exp(\ell_i - \ell_j)$;*
- *$\chi^2$-DRO: $q_i^*/q_j^* = (\ell_i - \lambda)_+/(\ell_j - \lambda)_+$, and $\lambda \geq 0$ is the dual parameter independent of $i$.*

Proposition 2.1 demonstrates that for KL-DRO, $\chi^2$-DRO, and GDRO (large gradient flow step), the *relative density* between samples is solely determined by their prediction errors, indicating that a larger prediction error results in a higher density. However, in our problem setting, samples from *both* minor sub-population $P_{minor}$ *and* noisy sub-population $\tilde{Q}$ exhibit high prediction errors. The primary goal of DRO is to focus on the minor sub-population $P_{minor}$, but the presence of noisy samples in $\tilde{Q}$ significantly interferes with this objective and hurts model learning. As shown in Figure 1, for KL-DRO, $\chi^2$-DRO and GDRO, noisy samples attract much density. Intuitively, it is not surprising that an excessive focus on noisy samples can have a detrimental impact. As KL-DRO, $\chi^2$-DRO, and

GDRO can be viewed as optimization within a weighted empirical distribution, we use the following simple example with the weighted least square model to demonstrate how this excessive focus on noisy samples can lead to high estimation variance, ultimately causing over-pessimism.

**Example** (Weighted Least Square). *Consider the data generation process as $Y = kX + \xi$, where $X, Y \in \mathbb{R}$ and random noise $\xi$ satisfies $\xi \perp X$, $\mathbb{E}[\xi] = 0$ and $\mathbb{E}[\xi^2]$ (abbr. $\sigma^2$) is finite. Assume that the training dataset $X_D$ consists of clean samples $\{x_c^{(i)}, y_c^{(i)}\}_{i \in [N_c]}$ and noisy samples $\{x_o^{(i)}, y_o^{(i)}\}_{i \in [N_o]}$ with $\sigma_c^2 < \sigma_o^2$. Consider the weighted least-square model $f(X) = \theta X$. Denote the sample weight of a clean sample $(x_c^{(i)}, y_c^{(i)})$ as $w_c^{(i)} \in \mathbb{R}_+, i \in [N_c]$, and the sample weight of a noisy sample $(x_o^{(i)}, y_o^{(i)})$ as $w_o^{(i)} \in \mathbb{R}_+, i \in [N_o]$ with $\sum_{i \in [N_c]} w_c^{(i)} + \sum_{i \in [N_o]} w_o^{(i)} = 1$. The variance of the estimator $\hat{\theta}$ is given by:*

$$Var[\hat{\theta}|X_D] = \frac{\sum_{i=1}^{N_c} (w_c^{(i)})^2 (x_c^{(i)})^2 \sigma_c^2 + \sum_{i=1}^{N_o} (w_o^{(i)})^2 (x_o^{(i)})^2 \sigma_o^2}{\left[\sum_{i=1}^{N_c} w_c^{(i)} (x_c^{(i)})^2 + \sum_{i=1}^{N_o} w_o^{(i)} (x_o^{(i)})^2\right]^2}, \tag{3}$$

*where $X_D = \{x_c^{(i)}\}_1^{N_c} \cup \{x_o^{(i)}\}_1^{N_o}$ are the sampled covariates in the dataset. Besides, the minimum variance is achieved if and only if $\forall 1 \le i \le N_c, 1 \le j \le N_o, w_o^{(j)}/w_c^{(i)} = \sigma_c^2/\sigma_o^2 < 1$.*

From the results, we make the following remarks:

• If noisy samples have higher weights than clean samples (e.g., $w_o/w_c > 1$), the variance of the estimated parameter $\hat{\theta}$ will be larger, suggesting that the learned $\hat{\theta}$ could be significantly unstable.

• In conjunction with Proposition 2.1, DRO methods tend to assign high weights to noisy samples, which can lead to unstable parameter estimation. While this example is relatively simple, this phenomenon aligns with the empirical findings in Zhai et al. (2021), which demonstrate that DRO methods can be quite unstable when confronted with label noise.

**Relationship with Conventional Outlier-robust Regression.** We would like to explain why conventional outlier-robust regression methods cannot be directly applied to our problem. The main challenge stems from the *coexistence* of noisy samples and minor sub-populations, both of which typically exhibit high prediction errors, leading to a misleading worst-case distribution in DRO. Conventional outlier-robust regression methods (Diakonikolas and Kane, 2018; Klivans et al., 2018; Diakonikolas et al., 2022) primarily focus on mitigating the effects of outliers without considering sub-population shifts. For instance, the $L_2$-estimation-error of outlier-robust linear regression is $\mathcal{O}(\epsilon \log(1/\epsilon))$ (Diakonikolas and Kane, 2018), where $\epsilon$ represents the noise level in Equation 1. However, as analyzed in Proposition 2.1 and demonstrated in Figure 1, during the optimization of DRO, the noise level $\epsilon$ significantly increases, rendering even outlier-robust estimation quite inaccurate. Moreover, Klivans et al. (2018) propose finding a pseudo distribution with minimal prediction errors to avoid outliers (see Algorithm 5.2 in (Klivans et al., 2018)). Nevertheless, this approach might inadvertently exclude minor sub-populations, which should be the focus under sub-population shifts, due to the main challenge: the *coexistence* of noisy samples and minor sub-populations. Zhai et al. (2021) incorporate this idea into DRO. Still, their method requires an implicit assumption that the prediction errors of noisy samples are higher than those of minor sub-populations, which does not always hold in practice. And Bennouna and Van Parys (2022) build the uncertainty set via two measures, KL-divergence and Wasserstein distance, leading to a combined approach of KL-DRO and ridge regression. Despite this, as we discussed earlier, DRO tends to increase the noise level in data, making it difficult to fix using ridge regression.

Based on the analysis above, we stress the importance of integrating more data-derived information. In pursuit of this, we propose to leverage the unique geometric properties that distinguish noisy samples from minor sub-populations to address this issue.

# 3 Proposed Method

In this work, with a focus on regression, we introduce our Geometry-Calibrated DRO (GCDRO). The fundamental idea is to utilize data geometry to distinguish between random noisy samples and minor sub-populations. It is motivated by the fact that prediction errors for minor sub-populations typically exhibit local smoothness along the data manifold, a property that is not shared by noisy samples.

**Discrete Geometric Wasserstein Distance.** We briefly revisit the definition of the discrete geometric Wasserstein distance. Given a weighted finite graph $G_N = (V, E, W)$, the probability set $\mathscr{P}(G_N)$ supported on the vertex set $V$ is defined as $\mathscr{P}(G_N) = \{\mathbf{p} \in \mathbb{R}^N \mid \sum_{i=1}^N p_i = 1, p_i \geq 0, \text{for } i \in V\}$, and its interior is denoted as $\mathscr{P}_o(G_N)$. A velocity field $\mathbf{v} = (v_{ij})_{i,j \in V} \in \mathbb{R}^{N \times N}$ on $G_N$ is defined on the edge set $E$ satisfying that $v_{ij} = -v_{ji}$ if $(i, j) \in E$. $\xi_{ij}(\mathbf{p})$ is a function interpolated with the associated nodes' densities $p_i, p_j$. The flux function $\mathbf{pv} \in \mathbb{R}^{N \times N}$ on $G_N$ is defined as $\mathbf{pv} := (v_{ij}\xi_{ij}(\mathbf{p}))_{(i,j) \in E}$ and its divergence is defined as $\text{div}_{G_N}(\mathbf{pv}) := -(\sum_{j \in V:(i,j) \in E} \sqrt{w_{ij}} v_{ij}\xi_{ij}(\mathbf{p}))_{i=1}^N \in \mathbb{R}^N$. Then for distributions $\mathbf{p}_0, \mathbf{p}_1 \in \mathscr{P}_o(G_N)$, the discrete geometric Wasserstein distance (Chow et al., 2017; Liu et al., 2022b) is defined as:

$$\mathcal{GW}_{G_N}^2(\mathbf{p}_0, \mathbf{p}_1) := \inf_v \left\{ \int_0^1 \frac{1}{2} \sum_{(i,j) \in E} \xi_{ij}(\mathbf{p}(t)) v_{ij}^2 dt \quad \text{s.t.} \frac{d\mathbf{p}}{dt} + \text{div}_{G_N}(\mathbf{pv}) = 0, \mathbf{p}(0) = \mathbf{p}_0, \mathbf{p}(1) = \mathbf{p}_1 \right\}. \tag{4}$$

Equation 4 computes the shortest (geodesic) length among all potential plans, integrating the total kinetic energy of the velocity field throughout the transportation process. A key distinction from the Wasserstein distance is that it only permits density to appear at the graph nodes.

**Formulation** Given training dataset $D_{tr} = \{(x_i, y_i)\}_{i=1}^N$ and a finite weighted graph $G_N = (V, E, W)$ representing the inherent structure of sample covariates. Denote the empirical marginal distribution as $\hat{P}_X$, the formulation of GCDRO is:

$$\min_{\theta \in \Theta} \underbrace{\sup_{\mathbf{q}:\mathcal{GW}_{G_N}^2(\hat{P}_X, \mathbf{q}) \leq \rho}}_{\text{Geometric Wasserstein set}} \left\{ \mathcal{R}_N(\theta, \mathbf{q}) := \sum_{i=1}^N q_i \ell(f_\theta(x_i), y_i) - \underbrace{\frac{\alpha}{2} \cdot \sum_{(i,j) \in E} w_{ij} q_i q_j (\ell_i - \ell_j)^2}_{\text{Calibration Term I}} - \underbrace{\beta \cdot \sum_{i=1}^N q_i \log q_i}_{\text{Calibration Term II}} \right\}, \tag{5}$$

where $\rho$ is the pre-defined radius of the uncertainty set, $\ell_i$ is the loss on the $i$-th sample and $w_{ij} \in W$ denotes the edge weight between sample $i$ and $j$. $\alpha$ and $\beta$ are hyper-parameters.

**Illustrations.** In our formulation, for any distribution $\mathbf{q}$ within the uncertainty set,
**Calibration term I** ($\sum_{(i,j) \in E} w_{ij} q_i q_j (\ell_i - \ell_j)^2$) calculates the *graph total variation* of prediction errors along the data manifold that is characterized by $G_N$. Intuitively, when *selecting the worst-case distribution*, this term imposes a penalty on distributions that allocate high densities to random noisy samples, as this allocation significantly amplifies the overall variation in prediction errors. Conversely, this term does not penalize distributions that allocate high densities to minor sub-populations, as their errors are smooth and have a relatively small impact on the total variation along the manifold. This differing phenomenon arises from the distinct geometric properties of random noisy samples and minor sub-populations, as samples from the latter typically cluster together on the data manifold. Further, *during the optimization of model parameter $\theta$*, this term acts like a variance term, resulting in a quantile-like risk objective, which helps to mitigate the effects of outliers.
**Calibration term II** ($\sum_{i=1}^N q_i \log q_i$) represents the negative entropy of distribution $\mathbf{q}$. As discussed in Section 3.2, during optimization, this term transforms into a non-linear *graph Laplacian operator* that encourages sample weights to be smooth along the manifold, avoiding extreme sample weights in the worst-case distribution.

## 3.1 Free Energy Implications on Worst-case Distribution

We first demonstrate the free energy implications of our risk objective $\mathcal{R}_N(\theta, \mathbf{q})$. Intuitively, the change of sample weights across $N$ samples (the inner maximization problem of $\mathcal{R}_N(\theta, \mathbf{q})$) can be analogously related to the dynamics of particles in a system, wherein the concentration of densities coincides with the aggregation of particle masses at $N$ distinct locations (in the case of infinite samples, these locations converge to the data manifold). As a result, a deeper understanding of the steady state in a particle system can offer *valuable insights into the worst-case distribution* for DRO.

Building on this analogy, we can dive deeper into the physics of particle interactions. When particles exist within a potential energy field, they are subject to external forces. Simultaneously, there are interactions among the particles themselves, leading to a constant state of motion within the system. In statistical physics, a key point of interest is identifying when a system reaches a steady state. In a standard process like the reversible isothermal process, it is established that spontaneous reactions consistently move in the direction of decreasing *Helmholtz free energy* (Fu et al., 1990; Reichl, 1999;

Friston, 2010), which consists of interaction energy, potential energy and the negative entropy:

$$\mathcal{E}(\mathbf{q}) = \underbrace{\mathbf{q}^\top K \mathbf{q}}_{\text{Interaction Energy}} + \underbrace{\mathbf{q}^\top V}_{\text{Potential Energy}} \underbrace{- \beta \sum_{i=1}^{N} (-q_i \log q_i)}_{\text{Temperature} \times \text{Entropy}} = -\mathcal{R}_N(\theta, \mathbf{q}). \qquad (6)$$

By taking $V = -\vec{\ell}$ and $K_{ij} = \frac{\alpha}{2} w_{ij} (\ell_i - \ell_j)^2$ for $(i,j) \in E$, our risk objective is a special case of Helmholtz free energy, where the potential energy of sample $i$ is $-\ell_i q_i$ and the interaction energy between sample $i$ and $j$ is $\frac{\alpha}{2} w_{ij} (\ell_i - \ell_j)^2 q_i q_j$. Specifically, such mutual interactions can manifest as *repulsive forces between adjacent particles*, thereby preventing the concentration of mass in locations where local prediction errors are significantly high. And this explains from a physical perspective why our calibration term **I** could mitigate random noisy samples.

Additionally, Proposition 3.1 offers physical interpretations to comprehend the worst-case distribution of various DRO methods. We make some remarks: (1) current DRO methodologies, except MMD-DRO, do not explicitly formulate the interaction term between samples in their design considerations ($\chi^2$-DRO does not involve interaction between samples), despite the corresponding interaction energy between particles being a common phenomenon in physics; (2) MMD-DRO simply uses kernel gram matrix for interaction and lacks efficient optimization algorithms; (3) by *considering this interaction energy*, our proposed GCDRO is capable of mitigating the impacts of random noisy samples.

**Proposition 3.1** (Free Energy Implications). *The dual reformulations of some typical DRO methods are equivalent to the free-energy-based minimax problem* $\min_{\theta \in \Theta, \lambda \geq 0} \max_{\mathbf{q} \in \mathscr{P}} \left\{ \lambda \rho - \mathcal{E}(\mathbf{q}, \theta, \lambda) \right\}$ *with different choices of* $\mathscr{P}, \rho$ *and* $K, V, H[q]$ *in the free energy* $\mathcal{E}$. *Details are shown in Table 1.*

Table 1: Free energy implications of some DRO methods. $\Delta_N$ denotes the $N$-dimensional simplex, $\eta$ in marginal DRO is the dual parameter.

| Method | Energy Type | | | Specific Formulation | | | |
|---|---|---|---|---|---|---|---|
| | Interaction | Potential | Entropy | $K$ | $V$ | $H[\mathbf{q}]$ | $\mathscr{P}$ |
| KL-DRO | ✗ | ✔ | ✔ | - | $-\vec{\ell}$ | $H[\mathbf{q}]$ | $\Delta_N$ |
| $\chi^2$-DRO | ✔ | ✔ | ✗ | $\lambda I$ | $-\vec{\ell}$ | - | $\Delta_N$ |
| MMD-DRO | ✔ | ✔ | ✗ | Kernel Gram Matrix $K$ | $-\vec{\ell} - \frac{2\lambda}{N} K^\top \mathbf{1}$ | - | $\Delta_N$ |
| Marginal $\chi^2$-DRO | ✗ | ✔ | ✗ | - | $-(\vec{\ell} - \eta)_+$ | - | $\Delta_N$ with Hölder continuity |
| GDRO | ✗ | ✔ | ✔ | - | $-\vec{\ell}$ | $H[\mathbf{q}]$ | Geometric Wasserstein Set |
| GCDRO | ✔ | ✔ | ✔ | Interaction Matrix $K$ | $-\vec{\ell}$ | $H[\mathbf{q}]$ | Geometric Wasserstein Set |

Through free energy, we could understand the type of energy or steady state that DRO methods strive to achieve, and design better interaction energy terms in DRO. Moreover, our optimization, as outlined in Section 3.2, could accommodate multiple quadratic forms of interaction energy.

## 3.2 Optimization

Then we derive an approximate minimax optimization for our GCDRO. For the *inner maximization* problem, we approximately deal with it via the gradient flow of $-\mathcal{R}_N(\theta, Q)$ w.r.t. $Q$ in the geometric Wasserstein space $(\mathscr{P}_o(G_N), \mathcal{GW}_{G_N})$. We show that the error rate is $\mathcal{O}(e^{-CT_{in}})$ after $T_{in}$ iterations inner loop, which gives a nice approximation. For the *outer minimization* w.r.t. model parameters $\theta$, we analyze the convergence rate of $\mathcal{O}(1/\sqrt{T_{out}})$ after $T_{out}$ iterations outer loop when the risk function satisfies Lipschitzian smoothness conditions.

**Inner Maximization.** We denote the *Continuous gradient flow* as $\mathbf{q}: [0, T] \to \mathscr{P}_o(G_N)$, the probability density of sample $i$ at time $t$ is abbreviated as $q_i(t)$, and the *Time-discretized gradient flow* with time step $\tau$ as $\hat{\mathbf{q}}_\tau$. For inner maximization, we utilize the $\tau$-time-discretized gradient flow (Villani, 2021) for $-\mathcal{R}_N(\theta, \mathbf{q})$ in the geometric Wasserstein space $(\mathscr{P}_o(G_N), \mathcal{GW}_{G_N}^2)$ as:

$$\hat{\mathbf{q}}_\tau(t + \tau) = \operatorname*{argmax}_{\mathbf{q} \in \mathscr{P}_o(G_N)} \mathcal{R}_N(\theta, \mathbf{q}) - \frac{1}{2\tau} \mathcal{GW}_{G_N}^2 (\hat{\mathbf{q}}_\tau(t), \mathbf{q}). \qquad (7)$$

The gradient of $\mathbf{q}$ in Equation 7 is given as (when $\tau \to 0$):

$$\frac{dq_i}{dt} = \sum_{(i,j) \in E} w_{ij} \xi_{ij} \left( \mathbf{q}, \ \ell_i - \ell_j + \beta(\log q_j - \log q_i) + \alpha \Big( \sum_{h \in N(j)} (\ell_h - \ell_j)^2 w_{jh} q_h - \sum_{h \in N(i)} (\ell_h - \ell_i)^2 w_{ih} q_h \Big) \right),$$

$$(8)$$

where $E$ is the edge set of $G_N$, $w_{ij}$ is the edge weight between node $i$ and $j$, $N(i)$ denotes the set of neighbors of node $i$, $\ell_i$ denotes the loss of sample $i$, and $\xi_{ij}(\cdot, \cdot) : \mathscr{P}(G_N) \times \mathbb{R} \to \mathbb{R}$ is:

$$\xi_{ij}(\mathbf{q}, v) := v \cdot \left( \mathbb{I}(v > 0)q_j + \mathbb{I}(v \leq 0)q_i \right), v \in \mathbb{R}, \tag{9}$$

which is the *upwind interpolation* commonly used in statistical physics and guarantees that the probability vector $\mathbf{q}$ keeps positive. From the gradient, we could see that the entropy regularization acts as a non-linear graph Laplacian operator to make the sample weights smooth along the manifold. In our algorithm, we fix the steps of the gradient flow to be $T_{in}$ and prove that the error ratio is $e^{-CT_{in}}$ compared with the *ground-truth* worst-case risk $\mathcal{R}_N(\theta, \mathbf{q}^*)$ constrained in an $\rho(\theta, T_{in})$-radius ball.

**Proposition 3.2** (Approximation Error Ratio). *Given the model parameter $\theta$, denote the distribution after time $T_{in}$ as $\mathbf{q}^{T_{in}}(\theta)$, and the distance to training distribution $\hat{P}_X$ as $\rho(\theta, T_{in}) := \mathcal{GW}_{G_N}^2(\hat{P}_X, \mathbf{q}^{T_{in}}(\theta))$ (abbr. $\rho(\theta)$). Assume $\mathcal{R}_N(\theta, \mathbf{q})$ is convex w.r.t $\mathbf{q}$. Then define the ground-truth worst-case distribution $q^*(\theta)$ within the $\rho(\theta)$-radius ball as:*

$$\mathbf{q}^*(\theta) := \arg \sup_{\mathbf{q} : \mathcal{GW}_{G_N}^2(\hat{P}_X, \mathbf{q}) \leq \rho(\theta)} \mathcal{R}_N(\theta, \mathbf{q}). \tag{10}$$

*The upper bound of the error rate of the objective function $\mathcal{R}_N(\theta, \mathbf{q}^{T_{in}})$ satisfies:*

$$(\mathcal{R}_N(\theta, \mathbf{q}^*) - \mathcal{R}_N(\theta, \mathbf{q}^{T_{in}}))/ \left( \mathcal{R}_N(\theta, \mathbf{q}^*) - \mathcal{R}_N(\theta, \hat{P}_X) \right) < e^{-CT_{in}}, \tag{11}$$

$$C = 2m\lambda_{sec}(\hat{L})\lambda_{min}(\nabla^2 \mathcal{R}_N)\frac{1}{(r+1)^2} > 0, \tag{12}$$

*where $\hat{L}$ is the Laplacian matrix of $G_N$. $\lambda_{sec}, \lambda_{min}$ are the second smallest and smallest eigenvalue, $m, r$ are constants depending on $\mathcal{R}_N, G_N, \beta$.*

We make some remarks:
- For the assumption that $\mathcal{R}_N$ is convex w.r.t. $\mathbf{q}$, the Hessian is given by $\nabla^2 \mathcal{R}_N = \beta \mathrm{diag}(1/q_1, ..., 1/q_N) + 2K$. Since $K$ is a sparse matrix whose nonzero elements in each row is far smaller than $N$, it is easily satisfied in empirical settings that the Hessian matrix $\nabla^2 \mathcal{R}$ is diagonally dominant and thus positive definite, making the inner maximization concave w.r.t $\mathbf{q}$.
- During the optimization, our algorithm finds an approximate worst-case distribution that is close to the ground-truth one within a $\rho(\theta)$-radius uncertainty set. Our robustness guarantee is similar to Sinha et al. (2018) (see Equation 12 in Sinha et al. (2018)).
- The error ratio is $e^{-CT_{in}}$, enabling to find a nice approximation efficiently with finite $T_{in}$ steps.

**Outer Minimization.** The convergence property relies on the risk objective $\mathcal{R}_N(\theta, \mathbf{q})$. When $\mathcal{R}_N(\theta, \mathbf{q})$ is *smooth* w.r.t. $\theta$, the following proposition guarantees convergence to a stationary point of problem 5 at a standard rate of $\mathcal{O}(1/\sqrt{T})$.

**Proposition 3.3** (Convergence). *Assume $F(\theta) := \sup_{\mathbf{q} : \mathcal{GW}_{G_N}^2(\hat{P}_X, \mathbf{q}) \leq \rho(\theta)} \mathcal{R}_N(\theta, \mathbf{q})$ is $L$-smooth, and $\mathcal{R}_N(\theta, \mathbf{q})$ is $L_q$-smooth w.r.t. $\mathbf{q}$ such that $\|\nabla_\mathbf{q} \mathcal{R}_N(\theta, \mathbf{q}) - \nabla_\mathbf{q} \mathcal{R}_N(\theta, \mathbf{q}')\|_2 \leq L_q \|\mathbf{q} - \mathbf{q}'\|_2$. $\rho(\theta)$ follows the definition in Proposition 3.2. Take a constant $\Delta_F \geq F(\theta^{(0)}) - \inf_\theta F(\theta)$ and set step size as $\alpha = \sqrt{\Delta_F/(LT)}$. For $\|\mathbf{q}^{T_{in}} - \mathbf{q}^*\|_2^2 \leq \gamma$ of the inner maximization problem, we have:*

$$\frac{1}{T}\mathbb{E}\left[\sum_{t=1}^T \|\nabla_\theta F(\theta^{(t)})\|_2^2\right] - \frac{1+2L\alpha}{1-2L\alpha}L_q^2 \gamma \leq \frac{2\Delta_F}{\sqrt{\Delta_F T} - 2L\Delta_F}. \tag{13}$$

From Proposition 3.3, as $T \to \infty$, $\nabla_\theta F(\theta^{(t)})$ exhibits a standard square-root convergence. Furthermore, the parameter $\gamma$ can be effectively controlled, owing to the concavity inherent in the inner maximization problem and the rapidly diminishing error ratio as described in Proposition 3.2.

## 3.3 Mitigate the Effects of Random Noisy Samples

Finally, we prove that our GCDRO method effectively de-emphasizes 'noisy samples' with locally non-smooth prediction errors. Due to the challenge of assessing intermediate states in gradient flow, we focus on its final state (as $T_{in} \to \infty$). Notably, in Proposition 3.2, the convergence rate of gradient flow is $\mathcal{O}(e^{-CT_{in}})$, implying that an efficient approximation of the final state is feasible.

For the worst-case distribution $q^*$, we denote the density ratio between samples as $\gamma(i, j) := q_i^*/q_j^*$. In sensitivity analysis, when *only* sample $i$ is perturbed with label noises, we denote the density ratio

in the new worst-case distribution $\tilde{q}^*$ as $\gamma^{\text{noisy}}(i,j) := \tilde{q}_i^*/\tilde{q}_j^*$. The sample weight sensitivity $\xi(i,j)$ is defined as $\xi(i,j) = \log \gamma^{\text{noisy}}(i,j) - \log \gamma(i,j)$, which measures how much density ratio changes under perturbations on one sample. Larger $\xi(i,j)$ indicates larger sensitivity to noisy samples.

**Proposition 3.4.** *Assume* $\ell_i^{noisy} - \ell_i \geq 2\left(\frac{\sum_{k \in N(i)} q_k^* w_{ik} \ell_k}{\sum_{k \in N(i)} q_k^* w_{ik}} - \ell_i\right)$ *which is locally non-smooth. For any* $\alpha > 0$ *(in Equation 5), we have* $\xi_{GCDRO} < \xi_{GDRO}$. *Furthermore, there exists* $M > 0$ *such that for any* $\alpha > M$, *we have* $\xi_{GCDRO}(i,j) < 0 < \min\{\xi_{\chi^2-DRO}(i,j), \xi_{GDRO}(i,j)(= \xi_{KL\text{-}DRO}(i,j))\}$, *indicating that GCDRO is not sensitive to locally non-smooth noisy samples.*

In practice, we do a grid search over $\alpha \in [0.1, 10]$ on an independent held-out validation dataset to select the best $\alpha$. The complexity of gradient flow scales *linearly* with sample size.

# 4  Experiments

In this section, we test the empirical performances of our proposed GCDRO on simulation data and real-world *regression* datasets with natural distributional shifts. As for the baselines, we compare with empirical risk minimization (ERM), WDRO, two typical $f$-DRO methods, including KL-DRO, $\chi^2$-DRO (Duchi and Namkoong, 2021), GDRO (Liu et al., 2022b), HRDRO (Bennouna and Van Parys, 2022) and DORO (Zhai et al., 2021), where HRDRO and DORO are designed to mitigate label noises.

Table 2: Results on the simulation data. We report the root mean square errors.

| | Weak Label Noise (noise level $0.5\%$) | | | | | Strong Label Noise (noise level $5\%$) | | | | |
|---|---|---|---|---|---|---|---|---|---|---|
| | Train (major) | Train (minor) | Test Mean | Test Std | Parameter Est Error | Train (major) | Train (minor) | Test Mean | Test Std | Parameter Est Error |
| ERM | 0.337 | 0.850 | 0.598 | 0.264 | 0.423 | 0.368 | 0.855 | 0.599 | 0.243 | 0.431 |
| WDRO | 0.337 | 0.851 | 0.589 | 0.292 | 0.424 | 0.368 | 0.857 | 0.600 | 0.268 | 0.432 |
| $\chi^2$-DRO | 0.596 | 0.765 | 0.680 | 0.088 | 0.447 | 1.072 | 0.708 | 0.875 | 0.193 | 0.443 |
| KL-DRO | 0.379 | 1.616 | 0.974 | 0.660 | 0.886 | 0.468 | 1.683 | 1.037 | 0.621 | 0.913 |
| HRDRO | 0.325 | 1.298 | 0.794 | 0.516 | 0.693 | 0.330 | 1.343 | 0.801 | 0.522 | 0.694 |
| DORO | 0.347 | 0.793 | 0.565 | 0.230 | 0.384 | 0.334 | 0.919 | 0.611 | 0.295 | 0.449 |
| GDRO | 0.692 | 0.516 | 0.605 | 0.094 | 0.198 | 0.618 | 0.752 | 0.677 | 0.063 | 0.421 |
| GCDRO | 0.411 | 0.554 | **0.482** | **0.070** | **0.190** | 0.494 | 0.591 | **0.540** | **0.044** | **0.268** |

## 4.1  Simulation Data

**Data Generation.**  We design simulation settings with both sub-population shifts and noisy samples. The input covariates $X = [S, U, V]^T \in \mathbb{R}^{10}$ consist of stable covariates $S \in \mathbb{R}^5$, irrelevant ones $U \in \mathbb{R}^4$ and the unstable covariate $V \in \mathbb{R}$:

$$[S, U] \sim \mathcal{N}(0, 2\mathbb{I}_9), Y = \theta_S^T S + 0.1 S_1 S_2 S_3 + \mathcal{N}(0, 0.5), V \sim \text{Laplace}(\text{sign}(r) \cdot Y, 1/5 \ln|r|), \quad (14)$$

where $\theta_S \in \mathbb{R}^5$ is the coefficients of the true model, $|r| > 1$ is the adjustment factor for each sub-population, and $\text{Laplace}(\cdot, \cdot)$ denotes the Laplace distribution. From the data generation, the relationship between $S$ and $Y$ stays invariant under different $r$, $U \perp Y$, while the relationship between $V$ and $Y$ is controlled by $r$, which *varies across sub-populations*. Intuitively, $\text{sign}(r)$ controls whether the spurious correlation $V$-$Y$ is positive or negative. And $|r|$ controls the strength of the spurious correlation: the larger $|r|$ is, the stronger the spurious correlation is. Furthermore, in order to conform to real data which are naturally assembled with label noises (Zhai et al., 2021), we introduce label noises by an $\epsilon$ proportion of labels as $Y' \sim \mathcal{N}(0, \text{Std}(Y))$. $\epsilon$ controls the noise level.
**Settings.**  In training, we generate 9,500 points with $r = 1.9$ (*majority*, strong positive spurious correlation $V$-$Y$) and 500 points with $r = -1.3$ (*minority*, weak negative spurious correlation $V$-$Y$). In testing, we vary $r \in \{3.0, 2.3, -1.9, -2.7\}$ to simulate different spurious correlations $V$-$Y$. We use *linear model* with mean square error (MSE) and report the prediction root-mean-square errors (RMSE) for each sub-population, the mean and standard deviation of prediction errors among all testing sub-populations. Also, we report the parameter estimation errors $\|\hat{\theta} - \theta^*\|_2$ of all methods ($\theta^* = (\theta_S^T, 0, \ldots, 0)^T$). The results over 10 runs are shown in Table 2.
 **Analysis.**  From Table 2, (1) compared with ERM, all typical DRO methods, especially $\chi^2$-DRO and KL-DRO, are strongly affected by label noises. (2) Although DORO is designed to mitigate outliers, it does not perform well under strong noises ($\kappa = 5\%$), because it relies on the assumption that noisy points have the largest prediction errors, which does not always hold. (3) Our proposed GCDRO outperforms all baselines under different strengths of label noises, which demonstrates its effectiveness. (4) Compared with GDRO, we could see that our *calibration terms* in Equation

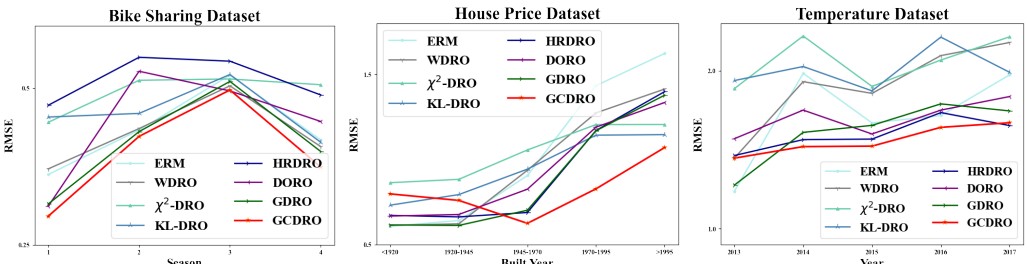

Figure 2: Results of real-world datasets with natural shifts. We do not manually add label noises here, since real-world datasets intrinsically contain noises.

5 is effective to mitigate label noises. From Figure 1, the worst-case distribution of our GCDRO *significantly upweighs on the minority* (green points) and does not put much density on the noisy data (red points), while the others put much higher weights on the noisy samples and perform poorly.

## 4.2 Real-world Data

We use three real-world regression datasets with natural distributional shifts, including bike-sharing prediction, house price, and temperature prediction. For all these experiments, we use a two-layer *MLP* model with mean square error (MSE). We use the Adam optimizer Kingma and Ba (2015) with the default learning rate $1e-3$. And all methods are trained for $5e3$ epochs.

**Datasets.** (1) **Bike-sharing** dataset (Dua and Graff, 2017) contains the daily count of rental bikes in the Capital bike-sharing system with the corresponding 11 weather and seasonal covariates. The task is to predict the count of rental bikes of *casual users*. Note that the count of casual users is likely to be more *random and noisy*, which is suitable to verify the effectiveness of our method. We split the dataset according to the season for natural shifts. In the training data, the ratio of four seasons' data is $9:7:5:3$. We test on the rest of the data and report the prediction error of each season. (2) **House Price** dataset[1] contains house sales prices from King County, USA. The task is to predict the transaction price of the house via 17 predictive covariates such as the number of bedrooms, square footage of the house, etc. We divide the data into 5 sub-populations according to the built year of each house with each sub-population covering a span of 25 years. In training, we use data from the first group (built year $< 1920$) and report the prediction error for each testing group. (3) **Temperature** dataset (Dua and Graff, 2017) is largely composed of the LDAPS model's next day's forecast data, in-situ maximum and minimum temperatures of present-day, and geographic auxiliary variables in South Korea from 2013 to 2017. The task is to predict the next-day's maximum air temperatures based on the 22 covariates. We divide the data into 5 groups corresponding with 5 years. In the training data, the ratio of five years' data is $9:7:5:3:1$. We test on the rest of the data and report the prediction error of each year. More details could be found in Appendix.

**Analysis.** (1) From the results in Figure 4.1, we could see that the performances of ERM drop a lot under distributional shifts, and DRO methods have better performance as well as robustness. (2) Our proposed GCDRO outperforms all baselines under strong shifts, with the most stable performances under natural distributional shifts. (3) As for the $k$NN graph's fitting accuracy of the data manifold, we visualize the learned manifold in Appendix and we could see that the learned $k$NN graph fits the data manifold well. Besides, we show in Appendix that the performances of our GCDRO are relatively stable across different choices of $k$. Also, our GCDRO only needs the input graph $G_N$ to represent the data structure and *any manifold learning or graph learning* methods could be plugged in to give a better estimation of $G_N$.

## 5 Future Directions

Our work deals with the over-pessimism in DRO via geometric calibration terms and provides free energy implications. The high-level idea could inspire future research on (1) relating free energy with DRO; (2) designing more reasonable calibration terms in DRO; (3) incorporating data geometry in general risk minimization algorithms.

---

[1]https://www.kaggle.com/c/house-prices-advanced-regression- techniques/data

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
