## A Implementation

For our GCDRO, $G_N$ is constructed as a $k$-nearest neighbor ($k$NN) graph from training data *once and for all **only at the initialization step***. For large-scale datasets, we use NN-Descent to estimate the $k$NN graph with an almost linear complexity of $\mathcal{O}(\mathbf{N}^{1.14})$. Since the sample weights are transferred along the edges of the graph, the simulation of gradient flow can be implemented similarly to message propagation with DGL package (Wang et al., 2019), which ***scales linearly with sample size*** and enjoys *parallelization by GPU*. The implementation above ensures the adaptability to large-scale data.

## B Improvements of our work.

In Section 2, we have introduced the typical DRO methods in detail and demonstrated the over-pessimism problem. Here we compare our work with several DRO works and clarify their differences. (1) With MMD-DRO: MMD-DRO (Staib and Jegelka, 2019) also has a quadratic term in its dual reformulation, while Staib and Jegelka (2019) focuses on the equivalence between MMD-DRO and Hilbert norms and there is no efficient or applicable algorithm yet. Further, it remains the risk objective unchanged (the quadratic term is from MMD distance) and just uses the Gaussian RBF kernel. Our work firstly incorporates the data geometry into the design of the calibration term and demonstrates its relationship with Helmholtz free energy, and we propose an applicable algorithm that could be used under deep models.
(2) With GDRO: GDRO (Liu et al., 2022b) uses the discrete geometric Wasserstein distance to build the uncertainty set, and intuitively demonstrates its superiority. Our work theoretically analyzes the over-pessimism problem and attributes the cause of over-pessimism to the excessive focus on noisy samples in DRO. And for the risk objective function, our work further introduces the graph total variation term to mitigate the effects of noisy samples, which is theoretically justified and empirically verified. From our results, GDRO is heavily affected by noisy samples, while our GCDRO has a much better performance. Further, this work relates the newly-proposed risk objective to the Helmholtz free energy and unifies some typical DRO methods into it, which is a new perspective to view DRO methods and could inspire future research.
(3) With DORO: DORO (Zhai et al., 2021) proposes to dismiss data samples with the top losses and then performs DRO, and we compare with it in our experiments. Theoretically, this method relies on the implicit assumption that noisy samples must have larger prediction errors than hard clean samples. However, this assumption does not always hold, and as shown in our experiments, it has some effects but does not work very well.

## C Why uses $k$NN graph?

**Manifold Assumption**. The data manifold hypothesis indicates that high-dimensional data often lies in an unknown lower-dimensional manifold embedded in ambient space Roweis and Saul (2000); Belkin and Niyogi (2003); Levina and Bickel (2004); Lunga et al. (2013); Brown et al. (2022) and is supported by strong evidence. From a theoretical perspective, Ozakin and Gray (2009); Narayanan and Mitter (2010) prove that when such hypothesis holds, manifold learning and density estimation scale exponentially with the *low intrinsic* dimension, but otherwise scale exponentially with the *high ambient* dimension Cacoullos (1964). Therefore, as Brown et al. (2022) point out, one most plausible explanation for the success of machine learning methods on real-world data is the existence of such lower intrinsic dimension, which enables learning on datasets of fairly reasonable size, which is empirically verified by Pope et al. (2021). Also, for two of the real-world tabular datasets used in this work, we visualize their 3-dimensional manifolds and calculate their intrinsic dimensions in Figure 3.

Our GCDRO algorithm uses an input-weighted graph $G_N$ to approximate the data manifold. The $k$NN graph is a fundamental and basic way to represent the data structure, and manifold learning is an area with intensive research. We have to clarify that manifold learning is not the focus of this paper, which takes the data structure $G_N$ as input to design a DRO objective and optimization algorithm that incorporates data geometric information for more reasonable worst-case distribution. Notably, our GCDRO achieves significant performance in the experiments even with the simple $k$NN representation of data structure. It proves that this direction for geometric-aware DROs is promising,

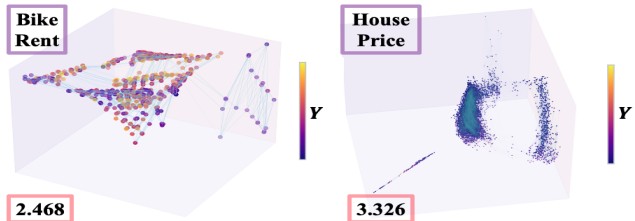

Figure 3: Visualization of the 3-dimensional manifold of the tabular datasets, and the numbers in the lower left represent the intrinsic dimension according to Levina and Bickel (2004)

and our proposed method could efficiently leverage the geometric properties encoded in the input graph to mitigate the effects of harmful data points (note that no target information is leaked into $G_N$). Actually, our GCDRO is compatible with any manifold learning or graph learning method. We do believe that a more accurate estimated data structure with advanced manifold learning algorithms will further boost the performance of GCDRO, and we leave this to future work.

**Not Sensitive to $k$.** For the house pricing dataset, we plot the results of our GCDRO with varying $k$s in Figure 4. We could see that the performance of our algorithm is not affected much.

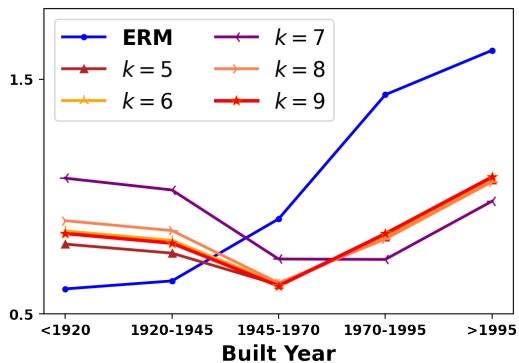

Figure 4: Results with varying $k$.

## D  Experimental Details

**Model & Loss function.** For simulation data, we use linear models for all methods. For real-world data, we use two-layer MLPs for all methods.

**Optimizer.** For all experiments, we use Adam with a learning rate of $1e - 3$ in PyTorch for all methods.

**Hyper-parameters.** For KLDRO, WDRO and $\chi^2$-DRO, we grid search the radius of the uncertainty set within the range of $[1e - 3, 2e2]$, and we select the best hyper-parameters according to their testing performances. For GDRO, we grid search the number of gradient flow steps within the range of $[1e2, 2e3]$, the parameter $\beta \in [1, 20]$ and we select the best hyper-parameters according to its testing performances. For DORO, we set the noisy ratio to the ground-truth value for the simulation data, and we grid search the ratio of noisy points within the range of $[1e - 2, 5e - 1]$ for the real-world data. For HRDRO, we use $L_1$ loss as proposed in (Bennouna and Van Parys, 2022) and grid search $\epsilon \in [1e - 3, 1]$. For GCDRO, we grid search the number of gradient flow steps within the range of $[1e2, 2e3]$, $\beta \in [1, 20]$ and $\alpha \in [1e - 1, 1e1]$. We select the best hyper-parameters according to their testing performances.

Note that in our experiments, we found that model selection without domain information in the validation set is very hard, which is also verified by Zhai et al. (2021); Gulrajani and Lopez-Paz (2021). And we believe this is still an open problem and is fairly non-trivial.

 # E    Examples on Label Noise

 **Theorem E.1.** *Assume that the training data is a mixture of $n_c$ clean samples $\{x_c^{(i)}, y_c^{(i)}\}$ drawn*
*from distribution $P_c(X, Y)$ and $n_o$ noisy samples $\{x_o^{(i)}, y_o^{(i)}\}$ drawn from distribution $P_o(X, Y)$.*
*Consider a linear data generation process, i.e. $Y = kX + \xi$ and $\xi \perp X, \mathbb{E}[\xi] = 0$ and $\mathbb{E}[\xi^2]$ is finite*
*(abbr. $\sigma^2$). The regression model is parameterized as $f(x) = \theta \cdot x$ and trained with Weighted Least*
*Square estimation:*

$$\hat{\theta} = \arg\min_{\theta} \sum_{i=1}^{n_c} w_c^{(i)} \|(y_c^{(i)} - \theta \cdot x_c^{(i)})\|^2 + \sum_{i=1}^{n_o} w_o^{(i)} \|(y_o^{(i)} - \theta \cdot x_o^{(i)})\|^2. \tag{15}$$

$$s.t. \ \ \sum_{i=1}^{n_c} w_c^{(i)} + \sum_{i=1}^{n_o} w_o^{(i)} = 1, \tag{16}$$

*where $w_c^{(i)}, w_o^{(i)} \geq 0$ are weights on clean and noisy samples respectively, and $\sigma_c^2 < \sigma_o^2$. Then the*
*variance of the least square estimate $\hat{\theta}$ is given by:*

$$Var[\hat{\theta}|X_D] = \frac{\sum_{i=1}^{n_c} (w_c^{(i)})^2 (x_c^{(i)})^2 \sigma_c^2 + \sum_{i=1}^{n_o} (w_o^{(i)})^2 (x_o^{(i)})^2 \sigma_o^2}{\left[ \sum_{i=1}^{n_c} w_c^{(i)} (x_c^{(i)})^2 + \sum_{i=1}^{n_o} w_o^{(i)} (x_o^{(i)})^2 \right]^2}, \tag{17}$$

*where $X_D = \{x_c^{(i)}\} \cup \{x_o^{(i)}\}$ is the sampled covariates in the dataset. Further, the variance of the*
*estimator $\hat{\theta}$ achieves the minimum if and only if:*

$$\forall 1 \leq i \leq n_c, 1 \leq j \leq n_o, \ \ \gamma(i, j) = w_o^{(j)} / w_c^{(i)} = \sigma_c^2 / \sigma_o^2, \tag{18}$$

*where $\gamma(i, j)$ denotes the sample weight ratio between $i$ and $j$.*

The theorem is a direct corollary of the following lemma.

**Lemma E.1.** *Assume that the training data contains $n$ samples $\{x^{(i)}, y^{(i)}\}$. Consider a linear data*
*generation process with heterogeneous noise, i.e. $y^{(i)} = kx^{(i)} + \xi_i$ with $\xi_i \perp X, \mathbb{E}[\xi_i] = 0$, and*
*$\mathbb{E}[\xi_i^2]$ is finite. The regression model is parameterized as $f(x) = \theta \cdot x$ and trained with Weighted*
*Least Square estimation:*

$$\hat{\theta} = \arg\min_{\theta} \sum_{i=1}^{n} w^{(i)} \|(y^{(i)} - \theta \cdot x^{(i)})\|^2. \tag{19}$$

$$s.t. \ \ \sum_{i=1}^{n} w^{(i)} = 1, \tag{20}$$

*where $w^{(i)} \geq 0$ are sample weights. Then the variance of the least square estimate $\hat{\theta}$ is given by:*

$$Var[\hat{\theta}|X_D] = \frac{\sum_{i=1}^{n} (w^{(i)})^2 (x^{(i)})^2 \sigma_i^2}{\left[ \sum_{i=1}^{n} w^{(i)} (x^{(i)})^2 \right]^2}, \tag{21}$$

*where $X_D = \{x^{(i)}\}$ is the sampled covariates in the dataset. Further, the variance of the estimator $\hat{\theta}$*
*achieves the minimum if and only if:*

$$\forall 1 \leq i \leq n, 1 \leq j \leq n, \ \ w^{(i)} \sigma_i^2 = w^{(j)} \sigma_j^2. \tag{22}$$

*Proof.* According to the heterogeneous noise distribution, let $y^{(i)} = x^{(i)} + \epsilon_i$, where $\epsilon_i \sim \mathcal{N}(0, \sigma_i^2)$.
The least square estimation of $\hat{\theta}$ is given by:

$$\hat{\theta} = k + \frac{\sum_{i=1}^{n} w^{(i)} x^{(i)} \epsilon_i}{\sum_{i=1}^{n} w^{(i)} (x^{(i)})^2}. \tag{23}$$

Since $\mathbb{E}[\hat{\theta}|X_D] = k$, we have

$$\text{Var}[\hat{\theta}|X_D] = \mathbb{E}\left|\frac{\sum_{i=1}^n w^{(i)} x^{(i)} \epsilon_i}{\sum_{i=1}^n w^{(i)} (x^{(i)})^2}\right|^2 \tag{24}$$

$$= \frac{\sum_{i=1}^n (w^{(i)})^2 (x^{(i)})^2 \sigma_i^2}{\left[\sum_{i=1}^n w^{(i)} (x^{(i)})^2\right]^2}. \tag{25}$$

Next, we solve the minimum of Eq.21 w.r.t. sample weights $w^{(i)}$. Let $\alpha_i = w^{(i)}(x^{(i)})^2$. We could formulate the variance in Eq.21 as a function of $\alpha = (\alpha_1, ..., \alpha_n)$:

$$V(\alpha) = \frac{\sum_{i=1}^n \alpha_i^2 \sigma_i^2/(x^{(i)})^2}{\left(\sum_{i=1}^n \alpha_i\right)^2}. \tag{26}$$

Since $V(\lambda\alpha) = V(\alpha)$ for any $\lambda > 0$, we could assume $\sum_{i=1}^n \alpha_i = 1$ without loss of generality. Then the minimization of $V(\alpha)$ is equivalent to:

$$\min_{\alpha} V(\alpha) = \sum_{i=1}^n \alpha_i^2 \sigma_i^2/(x^{(i)})^2. \tag{27}$$

$$s.t. \ \sum_{i=1}^n \alpha_i = 1. \tag{28}$$

The first-order KKT condition gives:

$$\exists C, \forall 1 \le i \le n, \ \alpha_i^* = C(x^{(i)})^2/\sigma_i^2, \tag{29}$$

from which we can solve:

$$\alpha_i^* = \frac{(x^{(i)})^2/\sigma_i^2}{\sum_{j=1}^n (x^{(j)})^2/\sigma_j^2}. \tag{30}$$

Since $\nabla_\alpha^2 V(\alpha) = diag\left[2\sigma_1^2/(x^{(1)})^2, ..., 2\sigma_n^2/(x^{(n)})^2\right]$ is always positive definite, Eq.30 minimizes $V(\alpha)$. Correspondingly $w^{(i)} \propto 1/\sigma_i^2$, which finishes the proof. $\qquad\square$

# F Proofs

## F.1 Proof of Proposition 2.1

*Proof.* (1) For KL-divergence as the distance function, we have the following optimization problem under finite samples.

$$\min_{\theta \in \Theta, \lambda \ge 0} \sup_{\mathbf{p} \in \Delta_n} \left\{ \sum_{i=1}^n p_i \ell(f_\theta(x_i), y_i) - \lambda \sum_{i=1}^n p_i \log p_i + \lambda(\epsilon - \log n) \right\}, \tag{31}$$

Solve the inner supremum problem, and the worst-case distribution is like:

$$p_i = \exp\left(\frac{\ell_i - \eta}{\lambda} - 1\right), \ \eta(\ell) = \lambda \log \lambda + \lambda \log\left(\sum_{i=1}^n \exp(\frac{\ell_i}{\lambda} - 1)\right), \tag{32}$$

and the objective function becomes:

$$\min_{\theta \in \Theta, \lambda \ge 0} \lambda \log\left(\sum_{i=1}^n \exp(\frac{\ell(f_\theta(x_i), y_i)}{\lambda})\right) + \lambda(\epsilon + \log \lambda - \log n). \tag{33}$$

And we could compare the sample weights of different samples as:

$$\frac{p_i}{p_j} = \exp(\frac{\ell_i - \ell_j}{\lambda}). \tag{34}$$

487 (2) For $\chi^2$-divergence which is defined as $f(x) = (x-1)^2$, we have the following optimization
488 problem.

$$\min_{\theta \in \Theta, \lambda \geq 0} \sup_{\mathbf{p} \in \Delta_n} \left\{ \sum_{i=1}^{n} p_i \ell(f_\theta(x_i), y_i) + \lambda\epsilon - \frac{\lambda}{n} \sum_{i=1}^{n} (np_i - 1)^2 \right\}. \tag{35}$$

489 Solve the inner supremum problem, and we have the worst-case distribution like:

$$p_i = \frac{1}{\lambda n}(\ell_i + \lambda - \eta)_+, \tag{36}$$

490 and the objective function becomes:

$$\min_{\theta \in \Theta, \lambda \geq 0, \eta \in \mathbb{R}} \sum_{i=1}^{n} \frac{1}{2\lambda}(\ell_i + \lambda - \eta)_+^2 + \lambda\epsilon + \eta - \frac{\lambda}{2}. \tag{37}$$

491 And we could compare the sample weights of different samples as:

$$\frac{p_i}{p_j} = \frac{(\ell_i + \lambda - \eta)_+}{(\ell_j + \lambda - \eta)_+}, \tag{38}$$

492 if $p_j > 0$.

493 (3) For Maximal Mean Discrepancy (MMD) distance, we have the following optimization problem:

$$\sup_{\mathbf{p}} \left\{ \sum_{i=1}^{n} p_i \ell_i + \lambda\epsilon - \lambda(\mathbf{p} - \frac{\mathbf{1}}{\mathbf{n}})^{\mathbf{T}} \mathbf{K}(\mathbf{p} - \frac{\mathbf{1}}{\mathbf{n}}) \right\} \tag{39}$$

$$\text{s.t.} \quad \sum_{i=1}^{n} p_i = 1 \tag{40}$$

$$p_i \geq 0, \text{for } i = 1, \ldots, n \tag{41}$$

494 Solve the inner supremum problem, and we have the worst-case distribution like:

$$p^* = \frac{1}{2\lambda} K^{-1}(\ell - \eta + \frac{2\lambda}{n}K\mathbf{1})_+, \tag{42}$$

495 and the objective function becomes:

$$\min_{\theta \in \Theta, \lambda \geq 0, \eta \in \mathbb{R}} \frac{1}{4\lambda}(\ell + \frac{2\lambda}{n}K\mathbf{1} - \eta)_+ K^{-1}(\ell + \frac{2\lambda}{n}K\mathbf{1} - \eta)_+ + \lambda\epsilon + \eta - \frac{\lambda}{n^2}\mathbf{1}^T K\mathbf{1}. \tag{43}$$

496 $\square$

## F.2 Proof of Proposition 3.1

498 *Proof.* Please refer to the proof of Proposition 2.1 for the proof of KL-DRO, $\chi^2$-DRO and MMD
499 DRO. For marginal DRO, it is easy to prove following Duchi et al. (2022). For GDRO, it is easy to
500 prove following Liu et al. (2022b). $\square$

## F.3 Proof of Proposition 3.2

502 *Proof.* The proof is based on the Theorem 5 in Chow et al. (2017). From Chow et al. (2017), we have

$$\mathcal{R}_N(q^\infty) - \mathcal{R}_N(q(t)) \leq e^{-Ct}(\mathcal{R}_N(q^\infty) - \mathcal{R}_N(q^0)). \tag{44}$$

503 Furthermore,

$$C := 2m\lambda_{\text{sec}}(\hat{L})\lambda_{\min}(\nabla^2 \mathcal{R}_N)\frac{1}{(r+1)^2} > 0, \tag{45}$$

504 and

$$r = \sqrt{2}k \max_{(i,j) \in E} w_{ij} \frac{\|\text{Hess}\mathcal{R}_N\|_1}{\lambda_{\min}(\text{Hess}\mathcal{R}_N)^{1.5}} \frac{1-m}{m^2} \frac{\lambda_{\max}(\hat{L})}{\lambda_{\text{sec}}(\hat{L})^2} \sqrt{\mathcal{R}_N(q^0) - \mathcal{R}_N(q^\infty)}, \tag{46}$$

where $k$ denotes the number of neighbors in the $k$NN graph, $\hat{L}$ is the graph Laplacian matrix, $\lambda_{\text{sec}}, \lambda_{\min}$ are the second smallest and smallest eigenvalue, and

$$\|\text{Hess}\mathcal{R}_N\|_1 = \sup_{q \in \mathscr{P}(G_N)} \|\text{Hess}\mathcal{R}_N(q)\|_1, \quad \lambda_{\min}(\text{Hess}\mathcal{R}_N) = \min_{q \in \mathscr{P}(G_N)} \lambda_{\min}(\text{Hess}\mathcal{R}_N(q)), \quad (47)$$

and

$$m = \frac{1}{2}\left(\frac{1}{(1+2M)^{\frac{1}{\beta}}}\right)^{N-2} \min\left\{\frac{1}{(1+2M)^{\frac{1}{\beta}}}\right), \frac{1}{N}\right\}. \quad (48)$$

Then denote the real worst-case distribution within the $\epsilon(\theta)$-radius discrete Geometric Wasserstein-ball as $q^*$, that is,

$$q^* = \arg \sup_{q:\mathcal{GW}^2_{G_N}(\hat{P}_{tr}, q) \leq \epsilon(\theta)} \mathcal{R}_N(\theta, q), \quad (49)$$

and we have

$$\mathcal{R}_N(q^\infty) - \mathcal{R}_N(q^*) + \mathcal{R}_N(q^*) - \mathcal{R}_N(q(t)) \leq e^{-Ct}(\mathcal{R}_N(q^\infty) - \mathcal{R}_N(q^*) + \mathcal{R}_N(q^*) - \mathcal{R}_N(q^0)). \quad (50)$$

Therefore, we have

$$\mathcal{R}_N(q^*) - \mathcal{R}_N(q(t)) \leq e^{-Ct}(\mathcal{R}_N(q^*) - \mathcal{R}_N(q^0)) - (1 - e^{-Ct})(\mathcal{R}_N(q^\infty) - \mathcal{R}_N(q^*)), \quad (51)$$

and

$$\frac{\mathcal{R}_N(q^*) - \mathcal{R}_N(q(t))}{\mathcal{R}_N(q^*) - \mathcal{R}_N(q^0)} \leq e^{-Ct} - (1 - e^{-Ct})\frac{\mathcal{R}_N(q^\infty) - \mathcal{R}_N(q^*)}{\mathcal{R}_N(q^*) - \mathcal{R}_N(q^0)} < e^{-Ct}. \quad (52)$$

$\square$

## F.4  Proof of Proposition 3.3

We follow Sinha et al. (2018) for proof of the convergence properties of our Algorithm. Denote the worst-case distribution as:

$$\mathbf{q}^*(\theta) = \arg \max_{\mathbf{q}:\mathcal{GW}^2_{G_0}(\hat{P}_X, \mathbf{q}) \leq \rho(\theta)} \mathcal{R}_N(\theta, \mathbf{q}), \quad (53)$$

and the outer-loop objective function as:

$$F(\theta) = \mathcal{R}_N(\theta, \mathbf{q}^*), \quad (54)$$

and our learned distribution after $k$ times gradient flow is denoted as $\mathbf{q}^k$. The gradient descent of $\theta$ is:

$$\theta^{(t+1)} = \theta^{(t)} - \alpha_t \cdot g^{(t)}. \quad (55)$$

By a Taylor expansion using the $L$-smoothness of the objective $F$, we have:

$$F(\theta^{(t+1)}) \leq F(\theta^{(t)}) + \langle \nabla_\theta F(\theta^{(t)}), \theta^{(t+1)} - \theta^{(t)} \rangle + \frac{L}{2}\|\theta^{(t+1)} - \theta^{(t)}\|_2^2 \quad (56)$$

$$= F(\theta^{(t)}) - \alpha \langle \nabla_\theta F(\theta^{(t)}), g^{(t)} \rangle + \frac{L\alpha^2}{2}\|g^{(t)}\|_2^2 \quad (57)$$

Then denote the gradient error as:

$$\delta^{(t)} = \nabla_\theta F(\theta^{(t)}) - g^{(t)} \quad (58)$$

thus, $g^{(t)} = \nabla_\theta F(\theta^{(t)}) - \delta^{(t)}$, and we have:

$$F(\theta^{(t+1)}) \leq F(\theta^{(t)}) - \alpha \langle \nabla_\theta F(\theta^{(t)}), g^{(t)} \rangle + \frac{L\alpha^2}{2}\|g^{(t)}\|_2^2 \quad (59)$$

$$\leq F(\theta^{(t)}) - \alpha \langle \nabla_\theta F(\theta^{(t)}), \nabla_\theta F(\theta^{(t)}) - \delta^{(t)} \rangle + \frac{L\alpha^2}{2}\|\nabla_\theta F(\theta^{(t)}) - \delta^{(t)}\|_2^2 \quad (60)$$

$$= F(\theta^{(t)}) - \alpha\|\nabla_\theta F(\theta^{(t)})\|_2^2 + \alpha \langle \nabla_\theta F(\theta^{(t)}), \delta^{(t)} \rangle + \frac{L\alpha^2}{2}\|\nabla_\theta F(\theta^{(t)}) - \delta^{(t)}\|_2^2 \quad (61)$$

$$\leq F(\theta^{(t)}) - \frac{\alpha}{2}\|\nabla_\theta F(\theta^{(t)})\|_2^2 + \frac{\alpha}{2}\|\delta^{(t)}\|_2^2 + L\alpha^2 \left(\|\nabla_\theta F(\theta^{(t)})\|_2^2 + \|\delta^{(t)}\|_2^2\right) \quad (62)$$

$$= F(\theta^{(t)}) - \frac{\alpha}{2}(1 - 2L\alpha)\|\nabla_\theta F(\theta^{(t)})\|_2^2 + \frac{\alpha}{2}(1 + 2L\alpha)\|\delta^{(t)}\|_2^2 \quad (63)$$

Therefore, we have:

$$(1 - 2L\alpha)\|\nabla_\theta F(\theta^{(t)})\|_2^2 - (1 + 2L\alpha)\|\delta^{(t)}\|_2^2 \le \frac{2}{\alpha}\left(F(\theta^{(t)}) - F(\theta^{(t+1)})\right) \tag{64}$$

and average from $t = 0$ to $K$ we have:

$$(1 - 2L\alpha)\frac{1}{K}\sum_{t=1}^{K}\|\nabla_\theta F(\theta^{(t)})\|_2^2 - (1 + 2L\alpha)\frac{1}{K}\sum_{t=1}^{K}\|\delta^{(t)}\|_2^2 \le \frac{2}{\alpha K}\left(F(\theta^{(0)}) - F(\theta^{(K+1)})\right) \tag{65}$$

and

$$\frac{1}{K}\sum_{t=1}^{K}\|\nabla_\theta F(\theta^{(t)})\|_2^2 - \frac{(1 + 2L\alpha)}{1 - 2L\alpha}\frac{1}{K}\sum_{t=1}^{K}\|\delta^{(t)}\|_2^2 \le \frac{2}{\alpha K(1 - 2L\alpha)}\left(F(\theta^{(0)}) - F(\theta^{(K+1)})\right) \tag{66}$$

Take expectations on the both sides like:

$$\frac{1}{K}\mathbb{E}\left[\sum_{t=1}^{K}\|\nabla_\theta F(\theta^{(t)})\|_2^2\right] - \frac{(1 + 2L\alpha)}{1 - 2L\alpha}\frac{1}{K}\sum_{t=1}^{K}\|\delta^{(t)}\|_2^2 \le \frac{2}{\alpha K(1 - 2L\alpha)}\left(F(\theta^{(0)}) - \mathbb{E}\left[F(\theta^{(K+1)})\right]\right) \tag{67}$$

Then we only have to bound the $\delta^{(t)}$. Following Sinha et al. (2018), we deal with $\|\delta^{(t)}\|_2^2$. According to the assumption on $\mathcal{R}_N(\theta, \mathbf{q})$, we have

$$\|\delta^{(t)}\|_2^2 = \|\nabla_\theta F(\theta^{(t)}) - g^{(t)}\|_2^2 = \|\nabla_{\mathbf{q}}\mathcal{R}_N(\theta^{(t)}, \mathbf{q}^*) - \nabla_{\mathbf{q}}\mathcal{R}_N(\theta^{(t)}, \mathbf{q}^{T_{in}})\|_2^2 \tag{68}$$

$$\le L_q^2\|\mathbf{q}^* - \mathbf{q}^{T_{in}}\|_2^2 \tag{69}$$

$$\le L_p^2\gamma \tag{70}$$

## F.5 Proof of Proposition 3.4

*Proof.* It is easy to prove that the final state of $\mathcal{R}_N(\theta, q)$ w.r.t. $q$ is given as

$$q_i^\infty = \frac{1}{Z}\exp\left(\frac{\ell_i - \alpha(\sum_{j \in N(i)} q_j^\infty w_{ij}(\ell_i - \ell_j)^2)}{\beta}\right), \tag{71}$$

where

$$Z = \sum_{i=1}^{N}\exp\left(\frac{\ell_i - \alpha(\sum_{j \in N(i)} q_j^\infty w_{ij}(\ell_i - \ell_j)^2)}{\beta}\right). \tag{72}$$

(1) When $\beta \to \infty$, $q_i^\infty \to \frac{1}{N}$. When $\beta \ll \infty$, the gradient flow is like:

$$\frac{dq_i}{dt} = \sum_{(i,j) \in E} w_{ij}\xi_{ij}\left(\ell_i - \ell_j + \beta(\log q_j - \log q_i) + \alpha\left(\sum_{h \in N(j)}(\ell_h - \ell_j)^2 w_{hj}q_h - \sum_{h \in N(i)}(\ell_h - \ell_i)^2 w_{hi}q_h\right)\right), \tag{73}$$

and

$$\xi_{ij}(v) := v \cdot \left(\mathbb{I}(v > 0)q_j + \mathbb{I}(v \le 0)q_i\right). \tag{74}$$

Therefore, when $q_i > q_j$ and $\ell_i > \ell_j$, we have $\log q_j - \log q_i < 0$, which decreases the gradient of $q_i$. Thus, the entropy term prompts the sample weights to be smooth between neighbors. When the sample weight of sample $i$ is larger than its neighbors, this term will decrease the gradient of $q_i$ to prevent it from gaining too much weights.

(3) Under the assumptions, we have

$$\left(\sum_{j \in N(i)} q_j^\infty w_{ij}(\ell_i + \Delta_i - \ell_j)^2 - \sum_{j \in N(i)} q_j^\infty w_{ij}(\ell_i - \ell_j)^2\right) \tag{75}$$

$$= \sum_{j \in N(i)} q_j^\infty w_{ij}(2\ell_i - 2\ell_j + \Delta_i)\Delta_i \tag{76}$$

$$\ge \Delta_i\left(\sum_{j \in N(i)} q_j^\infty w_{ij}(\Delta_i - 2L_x\|x_i - x_j\|_2)\right) \tag{77}$$

$$> \Delta_i. \tag{78}$$

539     Therefore, define $\delta_i = \ell_i^{\text{noisy}} - \ell_i$, it is easy to prove that for $\alpha > 0$

$$\xi_{\text{GCDRO}}(i, j) < \xi_{\text{GDRO}}(i, j), \tag{79}$$

540     and when $\alpha > \frac{1}{\sum_{k \in N(i)} q_k w_{ik}(2\ell_i - 2\ell_k + \delta_i)}$, we have $\xi_{\text{GCDRO}}(i, j) < 0$.     $\square$