# OpenReview forum: "Geometry-Calibrated DRO: Combating Over-Pessimism with Free Energy Implications"
_NeurIPS.cc/2023/Conference — Submitted to NeurIPS 2023_

### Official Review · Reviewer_WEic · 2023-07-05

**Soundness:** 3 good
**Presentation:** 3 good
**Contribution:** 3 good
**Rating:** 8
**Confidence:** 3

**Summary:**

The paper demonstrates a root cause of the over-pessimism issue of existing distributionally robust optimization (DRO) methods: excessive focus on noisy samples. To mitigate this issue, the authors proposed a novel DRO method called Geometrically-Calibrated DRO (GCDRO). They introduce the free energy implications of their method (Section 3.1) and approximate optimization method (Section 3.2). Finally, they empirically validate their method for both simulation data (Section 4.1) and real-world data (Section 4.2).

**Strengths:**

- While this paper is a theoretical work (for the ML community), I believe that the problems addressed in this paper can be applied in practical situations. Specifically, existing debiasing methods [1, 2] focus on minor group samples with loss/gradient information. Similar to existing DRO methods (as the paper shows), these debiasing methods will focus excessively on noisy samples. In this case, GCDRO (or a more computationally scalable approximation version) can be applied to mitigate this issue.
    - [1] Nam, Junhyun, et al. "Learning from failure: De-biasing classifier from biased classifier." *Advances in Neural Information Processing Systems* 33 (2020): 20673-20684.
    - [2] Ahn, Sumyeong, Seongyoon Kim, and Se-young Yun. "Mitigating Dataset Bias by Using Per-sample Gradient." *arXiv preprint arXiv:2205.15704* (2022).
- The free energy perspective of DRO using duality seems novel and interesting.

**Weaknesses:**

- The organization of this paper is somewhat puzzling and hard, although it deals with theoretical topics that are difficult for non-experts to understand. For example, it would be recommended to highlight the key difference between DRO and ERM in Section 2. As ERM does not assume graph structure G_N, readers unfamiliar with DRO would expect a detailed explanation/usage of the graph in L74.
- It would be recommended to add a confidence interval for the results in Figure 2. Also, a vector image format provides more clear results as it does not blur when zooming in.

**Questions:**

Can you provide the overall computation cost in terms of wall clock time? What is the computational bottleneck in the proposed method? Can the k-NN graph construction used in Appendix A be applied to debiasing settings in [1,2]?

**Limitations:**

The authors did not discuss the limitations of their method.

---

> ### Author Rebuttal · Authors · 2023-08-04
>
> We sincerely appreciate your approval of the theoretical and empirical contributions of this work. Thank you very much for the advice to improve this paper. And we address your concerns as follows:
>
> ### 1. Paper organization.
>
> Thanks for your suggestions. We will highlight the key difference between DRO and ERM in Section 2 in the camera-ready version. And as you suggest, we will demonstrate that the graph $G_N$ will be used as the input of our proposed method in Section 2 to avoid any confusion.
>
>
> ### 2. Figure 2.
>
> Thanks for your advice, and we would replace the figures with vector images (in .pdf format) to improve readability.
> As for the confidence interval, we provide the standard deviation of our proposed method's prediction errors on the house price dataset over 5 runs as follows:
>
> | GCDRO  |  $<1920$ | 1920~1945  | 1945~1970  | 1970~1995  | $>1995$|
> |---|---|---|---|---|---|
> | Mean  | 0.848  | 0.808  | 0.627  |  0.831 | 1.073|
> | Std  | 0.036  | 0.034  | 0.005  | 0.008  | 0.008 |
>
> From the results, we could observe that the standard deviations are relatively low. And we will add the error bars of all results to Figure 2 in the camera-ready version following your advice.
>
>
> ### 3. Computational burden
>
> There are two possible overheads of GCDRO compared with traditional DROs, such as WDRO.
> * First, a $k$-NN graph is constructed for each dataset respectively in our experiments. However, it is an initialization step and the effort is once and for all. We adopt NN-Descent [1] to construct the k-nearest neighbor graph with an almost linear complexity of $O(N^{1.14})$ for large-scale datasets. For small-scale datasets, we simply use the `scikit-learn` package to construct the $k$-NN graph.
> * Second, the inner maximization step of GCDRO is required to simulate the gradient flow. However, since the sample weights are only transferred along the edges of the graph, the simulation of gradient flow is implemented via the `DGL` package, a way similar to message propagation, which scales **linearly with sample size** and could be calculated via **GPU**. Thus, the computational cost of the inner maximization enjoys identical asymptotic time complexity as WDRO.
> * Finally, in empirical studies we found that a two-stage approximation of our algorithm could achieve very good results, which potentially further simplify the training procedure. This phenomenon is also observed in [2,3], and we leave it for future work to investigate in-depth the underlying reasons.
> * For the running time, we test our methods on the house price prediction dataset, which has 1305 training samples of 17 dimensions, the CPU time of building $k$-NN graph is 0.21s. The CPU time of GCDRO with a two-layer MLP model over 2000 epochs is 50.2s. And the CPU time of **one iteration of the gradient flow over $G_N$ (1305 nodes) is 0.003s**. Since the cost of gradient flow scales linearly with sample size $N$, this could show that the computational burden is relatively small. The experiment is run on the AMD EPYC 7402 24-Core Processor (only one core is used), and the GPU is NVIDIA GeForce RTX 3090.
>
> In summary, our method maintains a reasonable computational demand, making it tolerable for future large-scale applications. Moreover, it has been implemented to be compatible with GPU execution, further enhancing its accessibility and performance potential.
>
>
> [1] Hui Wang, Wan-Lei Zhao, Xiangxiang Zeng, Jianye Yang: Fast k-NN Graph Construction by GPU based NN-Descent. CIKM 2021.
>
> [2] Liu, E.Z., Haghgoo, B., Chen, A.S., Raghunathan, A., Koh, P.W., Sagawa, S., Liang, P. &amp; Finn, C.. Just Train Twice: Improving Group Robustness without Training Group Information. ICML 2021.
>
> [3] Idrissi, B.Y., Arjovsky, M., Pezeshki, M. &amp; Lopez-Paz, D.. Simple data balancing achieves competitive worst-group-accuracy. CleaR 2022.
>
>
> ### 4. Application for debiasing.
> Thanks for your recommended debiasing papers. We will cite them in our camera-ready version and provide more discussions.
> * There is potential to incorporate our methods in the debiasing setting. Here we only provide some intuitions and leave them for future exploration. Aftering training a biased model, we could use the gradient flow proposed in our work to learn the sample weights, which are likely to focus on minor subpopulation samples instead of noisy samples. And then a debiased model could be obtained via the sample weights. And our gradient flow process has strong physical implications and could be used in general settings (both classification and regression),
>
> * For the capacity of $k$-NN graph when dealing with complicated data, $k$-NN graph could be built upon the representations learned from the deep model. Similar techniques have been widely used in literature. In [1] and [2], the authors build a $k$-NN graph with representations learned for input samples and they test their methods on image datasets like CIFAR-10 and CIFAR-100. Therefore, we think $k$-NN graph could be used for complicated data. Also, some advanced manifold learning methods could be further incorporated to get better estimations.
>
> [1] Pengxiang Wu, Songzhu Zheng, Mayank Goswami, Dimitris N. Metaxas, Chao Chen: A Topological Filter for Learning with Label Noise. NeurIPS 2020.
>
> [2] Ahmet Iscen, Jack Valmadre, Anurag Arnab, Cordelia Schmid: Learning with Neighbor Consistency for Noisy Labels. CVPR 2022.

---

> > ### Comment · Reviewer_WEic · 2023-08-16
> >
> > Thank you for your detailed explanation and additional experiments. I updated my score to believe the following will be reflected in your revised manuscript (or camera-ready version).
> >
> > * The confidence interval would be included in all results of Figure 2.
> > * The discussion on computational complexity (including a proper reference to NN-Descent) would be included in the main text (or the Appendix with a clear reference in the main text).

---

> > > ### Author Response · Authors · 2023-08-17
> > > **Thanks for your support**
> > >
> > > Thank you for your approval of this work. Thanks for your suggestions, and we will include the confidence interval and computational complexity in the camera-ready version.

---

### Official Review · Reviewer_DBQo · 2023-07-07

**Soundness:** 3 good
**Presentation:** 3 good
**Contribution:** 3 good
**Rating:** 7
**Confidence:** 1

**Summary:**

In this work, the authors propose a novel approach called Geometry-Calibrated Distributionally Robust Optimization (GCDRO) to address the over-pessimism issue in traditional Distributionally Robust Optimization (DRO) methods. DRO aims to optimize worst-case risk within an uncertainty set to mitigate the effects of distributional shifts in machine learning algorithms. However, DRO often leads to low-confidence predictions, poor parameter estimations, and limited generalization.

The authors analyze a possible cause of over-pessimism in DRO, which is the excessive focus on noisy samples. To mitigate the impact of noise, they incorporate data geometry into calibration terms in DRO, resulting in GCDRO specifically designed for regression tasks. They demonstrate that their risk objective aligns with the concept of Helmholtz free energy in statistical physics, and this free-energy-based risk can be extended to standard DRO methods.

To optimize the GCDRO objective, the authors leverage gradient flow in the Wasserstein space and develop an approximate minimax optimization algorithm. This algorithm guarantees a bounded error ratio and standard convergence rate. The authors further explain how their approach alleviates the effects of noisy samples in the optimization process.

To validate the effectiveness of GCDRO, the authors conduct comprehensive experiments. These experiments demonstrate that GCDRO outperforms conventional DRO methods in terms of prediction accuracy, parameter estimation, and generalization performance.

Overall, the proposed GCDRO approach addresses the over-pessimism issue in DRO by incorporating data geometry into the calibration terms. The authors provide a theoretical analysis and empirical evidence to support the superiority of GCDRO over conventional DRO methods, showing its potential for improving the robustness and performance of machine learning algorithms in the face of distributional shifts and noisy samples.

**Strengths:**

Novel Approach: The paper introduces a novel approach called Geometry-Calibrated Distributionally Robust Optimization (GCDRO) to address the over-pessimism issue in traditional Distributionally Robust Optimization (DRO) methods. This new approach incorporates data geometry into calibration terms, offering a unique perspective on mitigating the impact of noisy samples and improving the performance of machine learning algorithms.

Theoretical Analysis: The paper provides a theoretical analysis of the over-pessimism issue in DRO and establishes a connection between the risk objective in GCDRO and the Helmholtz free energy in statistical physics. This theoretical analysis deepens the understanding of the problem and the proposed solution, providing a solid foundation for the proposed method.

Optimization Algorithm: The paper leverages gradient flow in Wasserstein space to develop an approximate minimax optimization algorithm for GCDRO. This algorithm guarantees a bounded error ratio and standard convergence rate, providing a reliable and efficient optimization framework for GCDRO.

Empirical Evaluation: The paper includes comprehensive experiments to evaluate the performance of GCDRO compared to conventional DRO methods. The experimental results demonstrate the superiority of GCDRO in terms of prediction accuracy, parameter estimation, and generalization performance. The empirical evaluation strengthens the claims made in the paper and highlights the practical benefits of the proposed approach.

Practical Significance: The paper addresses an important problem in machine learning—dealing with distributional shifts and noisy samples—and offers a practical solution that can improve the robustness and generalization of machine learning algorithms. The proposed GCDRO method has the potential to be applied in real-world scenarios where distributional shifts are prevalent, making it highly relevant and valuable.

**Weaknesses:**

Limited Comparison: The paper compares GCDRO only with conventional DRO methods, without exploring a broader range of state-of-the-art approaches or alternative methods for addressing the over-pessimism issue in DRO. Including a more diverse set of baselines would provide a more comprehensive evaluation and better contextualize the performance of GCDRO.

Lack of Real-world Applications: The paper focuses on theoretical analysis and empirical evaluations using synthetic or benchmark datasets. However, the absence of real-world applications or case studies limits the understanding of how GCDRO would perform in practical scenarios. Extending the evaluation to real-world datasets and applications would enhance the applicability and relevance of the proposed method.



**Questions:**

None

**Limitations:**

See weaknesses

---

> ### Author Rebuttal · Authors · 2023-08-04
>
> Thanks for your time and efforts in the reviewing process and we appreciate your approval of our methodological novelty, theoretical soundness, and empirical significance.
>
> * For baselines, we have compared most of the typical DRO methods, including two $\phi$-divergence DROs: KL-DRO and $\chi^2$-DRO, Wasserstein DRO, and Geometric DRO. We also compare HRDRO and DORO, which are designed to mitigate label noises in DRO.
> * For Real-world Applications, we test our proposed method on three real-world tabular datasets, which could validate the effectiveness of our method.
>
> Thanks for your suggestions and we will explore more applications of our method in the future.

---

### Official Review · Reviewer_rKdP · 2023-07-11

**Soundness:** 3 good
**Presentation:** 3 good
**Contribution:** 3 good
**Rating:** 6
**Confidence:** 3

**Summary:**

This work investigates the impact of noisy samples on Distributionally Robust Optimization (DRO) algorithms. Using a simple model and empirically, they show how DRO variants tend to give to much importance to those samples, which make them overly pessimistic in finding the worst case distribution shift. To solve this problem they propose a new method called Geometrically Calibrated DRO (GCDRO), which adds two new calibration terms. One of this term is looking at the relationship between samples (provided by a graph $G_N$) and penalizes shifted distributions $\mathbf{q}$ which assign high probability mass to connected samples when those have very different prediction losses.The other entropy term favors shifted distributions $\mathbf{q}$ with larger entropy, penalizing $\mathbf{q}$ which overly focus on a subset of samples. The authors provide an algorithm to approximately solve the inner maximization loop which they use to apply their algorithms to several benchmarks in which GCDRO is outperforming prior methods. Moreover, the authors draw a parallel between their method and Helmholtz free energy.

**Strengths:**

Originality: While the two added calibration terms are not entirely novel on their own, I find some novelty in combining them and (i) demonstrating a parallel with Helmholtz free energy, as well as (ii) showing how those can tackle the specific problem of noisy samples in DRO.

Clarity: Albeit dense, the paper is clear enough. I would maybe recommend clarifying early on how to get the graph $G_N$.

Quality & significance: This work is well motivated and I can see it being impactful. Especially, the link between DRO methods and statistical physics could motivate interesting future directions.

**Weaknesses:**

The method requires a graph $G_N$, which might not be easy to get. There should be a paragraph on how this graph was obtained for each dataset in the experiments section. In the supplementary, it is mentioned that $k$-NN graphs are used and the authors show that the results do not depends so much on $k$. While $k$-NN graphs might work on simple regression datasets, I doubt they would be efficient on more complex ones. For instance, DORO has been applied to CelebA, I'm not sure a $k$-NN graph would work there. Some more complex regression datasets can be found as part of WILDS [1]. I understand the goal of the paper is not manifold learning, yet better understanding the limitations associated with requiring $G_N$ seems important to grasp how useful the proposed method can be in practice.

Would proposition 3.3 still hold considering the error obtained from the inner maximization?

References:

[1] WILDS: A Benchmark of in-the-Wild Distribution Shifts (https://www-cs-faculty.stanford.edu/people/jure/pubs/wilds-icml21.pdf)

**Questions:**

- Would proposition 3.3 still hold considering the error obtained from the inner maximization?

**Limitations:**

The limitations of the method are somewhat discussed in the supplementary. I like that the authors did experiments with several $k$-NN graphs, but I believe this analysis can be more convincing by testing the method on more complex regression datasets.

---

> ### Author Rebuttal · Authors · 2023-08-04
>
> We sincerely appreciate your approval of the motivation, novelty, and potential impacts of this work.
> Thank you very much for the advice to improve this paper. And we address your concerns as follows:
>
> ### 1. How $G_N$ is obtained:
> We construct a $k$-NN graph using the raw covariates for each dataset respectively. The default $k$ is 5. We use the `scikit-learn` package in our experiments. Thanks for your suggestions and we will add details in Sections 4.1 and 4.2 to clarify.
>
> ### 2. $k$-NN for complex data:
> Thanks for your suggestions to clarify this. We will add an analysis of the computational burden as well as the limitations in the final version.
>
> 1. For complicated data (like image): $k$-NN graph could be built upon the representations learned from the deep model. Similar techniques have been widely used in literature. In [1] and [2], the authors build a $k$-NN graph with representations learned for input samples and they test their methods on image datasets like CIFAR-10 and CIFAR-100. Therefore, we think $k$-NN graph could be used for complicated data. Also, as the reviewer points out, some manifold learning methods could be further incorporated to get better estimations. This is not the focus of this work, and we leave it for future work.
> 2. For efficiency of $k$-NN construction: first, we argue that it is an initialization step and the effort is once and for all. Second, when the sample size is large, we use the NN-Descent [3] to construct the k-nearest neighbor graph with an almost linear complexity of $O(N^{1.14})$ for large-scale datasets. Furthermore, since the sample weights are transferred along the edges of the graph, the simulation of gradient flow can be implemented in a way similar to message propagation, which scales linearly with sample size.
> 3. Limitations associated with requiring $G_N$: As indicated in [1,2], $k$-NN graph could deal with complicated data (like images). Actually, our GCDRO is compatible with any manifold learning or graph learning method. We believe that a more accurate estimated data structure with advanced manifold learning algorithms will further boost the performance of GCDRO, and we leave this to future work. Besides, we think that there is huge potential to apply our method to graph data, where $G_N$ is pre-defined and could be directly used. But this is not the main focus of this work and we leave this for future explorations.
>
> We will move this part from the appendix to the main body in the camera-ready version.
>
> [1] Pengxiang Wu, Songzhu Zheng, Mayank Goswami, Dimitris N. Metaxas, Chao Chen: A Topological Filter for Learning with Label Noise. NeurIPS 2020.
>
> [2] Ahmet Iscen, Jack Valmadre, Anurag Arnab, Cordelia Schmid: Learning with Neighbor Consistency for Noisy Labels. CVPR 2022.
>
> [3] Hui Wang, Wan-Lei Zhao, Xiangxiang Zeng, Jianye Yang: Fast k-NN Graph Construction by GPU based NN-Descent. CIKM 2021.
>
>
>
> ### 3. Proposition 3.3 takes into consideration the error obtained from inner maximization.
>
> Proposition 3.3 has already taken into consideration the error obtained from the inner maximization. Specifically, we assume that the weights obtained from the inner loop have a bounded distance from the true weights such that $\|q^{T_{in}}-q^*\|\leq \gamma$. And it is shown in proposition 3.3 that the error rate $\gamma$ has a fixed effect on optimization accuracy, independent of $T$. It proves that approximation in the inner loop has limited propagation. Such characterization of approximate inner-level optimization is standard in DRO literature, as shown in [1].
>
> [1] Aman Sinha, Hongseok Namkoong, John C. Duchi: Certifying Some Distributional Robustness with Principled Adversarial Training. ICLR 2018.

---

> > ### Comment · Reviewer_rKdP · 2023-08-21
> > **Thank you for your rebuttal**
> >
> > I thank you for your rebuttal. My concerns are mostly well addressed and I decide to keep my score.

---

### Official Review · Reviewer_aeCE · 2023-07-16

**Soundness:** 3 good
**Presentation:** 3 good
**Contribution:** 2 fair
**Rating:** 5
**Confidence:** 5

**Summary:**

This paper introduces a novel approach to tackle the issue of overconservativeness in conventional DRO models. The proposed method incorporates data geometry properties into the design of the objective function and ambiguity set. The authors leverage the discrete geometric Wasserstein distance, initially presented in Chow et al. (2017), as a probability metric to construct the ambiguity set. Additionally, they enhance the model's performance by introducing the graph total variation quantity to the objective function. This modification effectively diminishes the influence of detrimental data points, thus mitigating their impact on the overall optimization process.
The effectiveness of the proposed approach is extensively validated through experiments conducted on synthetic and real datasets. The results demonstrate its superiority over conventional DRO models, exhibiting improved accuracy and robustness.

**Strengths:**

I acknowledge the significance of addressing the challenge posed by noisy examples in machine learning, and I commend the motivation behind the design presented in this paper. The proposed method is intriguing, and the introduction of new calibration terms appears to be a reasonable approach. The experimental results showcase promise, particularly on synthetic datasets, and also demonstrate competitiveness on real datasets.

**Weaknesses:**

1.The Discrete Geometric Wasserstein Distance restricts the worst-case distribution to have the same support as the training dataset. If this is the case, then the method you propose is fundamentally distinct from Wasserstein DRO. I am interested in understanding the specific scenarios in which Wasserstein DRO outperforms divergence-based methods. As far as I comprehend, GCDRO aims to enhance phi-divergence DRO methods by incorporating graph information.
2. I kindly request verification of the assumption made in Proposition 3.3, which states that F(\theta) is L-smooth. It appears that the convergence results were obtained based on this assumption alone.

**Questions:**

See weakness

---

> ### Author Rebuttal · Authors · 2023-08-04
>
> We sincerely appreciate your approval of the motivation and novelty of this work. Thank you very much for the advice to improve this paper. And we address your concerns as follows:
>
>
> ### 1. Comparison between WDRO and GCDRO
>
> * **when WDRO is better**:
> WDRO could extend the distribution support and has out-of-sample generalization guarantees. It relates to some traditional robustness penalties for linear models. For complicated data or deep models, [1] demonstrates its connection to adversarial training. We think that WDRO is more advantageous over $\phi$-divergence DRO in terms of robustness against adversarial attacks since it adds adversarial perturbations to data samples during training.
>
> * **when $\phi$-divergence based DRO is better**:
> Compared with WDRO, $\phi$-divergence based DRO restricts the worst-case distribution to have the same support as the training dataset and the worst-case distribution is characterized via sample weights. We think it better mitigates subpopulation shifts, where there are typically a major and a minor subpopulation. Reweighting could be better than generating adversarial samples to enhance the model's focus on the minor subpopulation by rebalancing the sample weights. With high-dimensional data, generating samples is hard in practice (maybe easier with recent generative models). Specifically, assume that a sample consists of covariates $X$ and target $Y$. Subtle perturbations are only exerted on the covariates $X$, implying a primary hidden assumption that target labels $Y$ remain invariant under subtle perturbations on covariates. Thus, adversarial training cannot address **general distributional shift** problems such as covariate shifts and concept drifts because these types of shifts require new $(X, Y)$ samples where perturbations of $X$ propagate to $Y$.
>
> * **GCDRO with WDRO**
> In this work, we focus on the setting where both minor subpopulation and label noises exist, and we do not focus much on adversarial robustness.
> We argue that GCDRO is also connected to WDRO. The geometric Wasserstein distance also measures the optimal transport cost of the worst-case distribution. The difference is that the classic Wasserstein distance transports mass along a straight line in the Euclidean space while the geometric Wasserstein distance transports mass along the manifold (edges of the graph). Thus, GCDRO could also be viewed as an enhanced version of WDRO which takes into consideration the manifold assumption.
>
>
> ### 2. $L$-smooth assumption.
> Throughout our experiments, we use the mean square error, but we admit that the smooth condition is hard to analyze with neural networks. Besides, this is a common assumption to analyze the convergence, as done in [1].
>
> [1] Aman Sinha, Hongseok Namkoong, John C. Duchi: Certifying Some Distributional Robustness with Principled Adversarial Training. ICLR 2018.

---

### Decision · Program_Chairs · 2023-09-21

**Decision:**

Reject

**Comment:**

The paper has received overall favorable reviews, with the reviewers appreciating its novelty and potential impact.

However, Reviewer aeCE has raised a concern over the smoothness assumption in Proposition 3.3 that was not properly addressed.  Even for quadratic losses and linear predictors the function $\mathcal{R}_N$ will not be smooth in $\theta$ due to Calibration Term I, and the maximization in the definition of $F$ further reduces smoothness (though Calibration Term II helps here). Moreover, the assumption that $\mathcal{R}_N$ is smooth in $q$ is also false, since entropy is not smooth. In contrast to claims by the authors, these smoothness assumptions are very different from the one in Sinha et al., which assumes smoothness of the loss with respect to the input examples.

Granted, Proposition 3.3 is not the core of the paper, but the issues above reduce our confidence in the overall theoretical soundness of the paper. Moreover, the experiments alone do not provide sufficient evidence for the proposed method’s efficacy, as they do not improve performance obtained by prior work on well-established settings such as WILDS.

In follow-up discussion the reviewers agreed on the two considerations above, still expressing support for the paper but having low confidence in its claims. Consequently, I believe the paper needs a major revision improving its theoretical rigor and perhaps also strengthening its experiments.